# Defined neuronal populations drive fatal phenotype in a mouse model of Leigh syndrome

Irene Bolea[1,2†]*, Alejandro Gella[2,3†], Elisenda Sanz[2,4,5†], Patricia Prada-Dacasa[2,5], Fabien Menardy[2], Angela M Bard[4], Pablo Machuca-Márquez[2], Abel Eraso-Pichot[2], Guillem Mòdol-Caballero[2,5,6], Xavier Navarro[2,5,6], Franck Kalume[4,7,8], Albert Quintana[1,2,4,5,9‡]*

[1]Center for Developmental Therapeutics, Seattle Children's Research Institute, Seattle, United States; [2]Institut de Neurociències, Universitat Autònoma de Barcelona, Bellaterra, Spain; [3]Department of Biochemistry and Molecular Biology, Universitat Autònoma de Barcelona, Bellaterra, Spain; [4]Center for Integrative Brain Research, Seattle Children's Research Institute, Seattle, United States; [5]Department of Cell Biology, Physiology and Immunology, Universitat Autònoma de Barcelona, Bellaterra, Spain; [6]Centro de Investigación Biomédica en Red sobre Enfermedades Neurodegenerativas (CIBERNED), Bellaterra, Spain; [7]Department of Neurological Surgery, University of Washington, Seattle, United States; [8]Department of Pharmacology, University of Washington, Seattle, United States; [9]Department of Pediatrics, University of Washington, Seattle, United States

*For correspondence:
irene.bolea@uab.cat (IB);
albert.quintana@uab.cat (AQ)

†These authors contributed equally to this work

Present address: ‡Institut de Neurociències, Universitat Autònoma de Barcelona, Bellaterra (Barcelona), Spain

Competing interests: The authors declare that no competing interests exist.

**Abstract** Mitochondrial deficits in energy production cause untreatable and fatal pathologies known as mitochondrial disease (MD). Central nervous system affectation is critical in Leigh Syndrome (LS), a common MD presentation, leading to motor and respiratory deficits, seizures and premature death. However, only specific neuronal populations are affected. Furthermore, their molecular identity and their contribution to the disease remains unknown. Here, using a mouse model of LS lacking the mitochondrial complex I subunit *Ndufs4*, we dissect the critical role of genetically-defined neuronal populations in LS progression. *Ndufs4* inactivation in Vglut2-expressing glutamatergic neurons leads to decreased neuronal firing, brainstem inflammation, motor and respiratory deficits, and early death. In contrast, *Ndufs4* deletion in GABAergic neurons causes basal ganglia inflammation without motor or respiratory involvement, but accompanied by hypothermia and severe epileptic seizures preceding death. These results provide novel insight in the cell type-specific contribution to the pathology, dissecting the underlying cellular mechanisms of MD.
DOI: https://doi.org/10.7554/eLife.47163.001

## Introduction

Leigh syndrome (LS) is the most frequent pediatric mitochondrial disorder, leading to defective mitochondrial energy metabolism. LS affects 1 in 40,000 births (*Rahman et al., 1996*), although adult onset has also been described (*McKelvie et al., 2012*). Mutations in more than 75 genes have been described to cause LS (*Lake et al., 2016*). To date, no effective treatment or cure exists. Clinically, albeit highly variable, LS symptoms usually include failure to thrive, hypotonia, rigidity, seizures, ataxia, lactic acidosis, encephalopathy and premature death (*Rahman et al., 1996*; *Lake et al., 2016*; *Sofou et al., 2014*). LS is characterized by restricted anatomical and cellular

**eLife digest** Mitochondria are often described as the power plants of cells because they generate most of the energy that a cell needs to survive. But one in every 5,000 children is born with a mutation that leads to faulty mitochondria, which generate less energy than their healthy counterparts. This is particularly problematic for tissues with high energy demands, such as the brain and muscles. Children with such mutations are said to have mitochondrial disease, and one of the most common and severe forms is Leigh syndrome.

Children with Leigh syndrome suffer from epilepsy, and have difficulties with movement and breathing. There is no treatment for Leigh syndrome, and most of those affected will die in childhood. The brains of children with Leigh syndrome show a characteristic pattern of damage and inflammation, symmetrical across both hemispheres, with two areas of the brain affected the most. First, the brainstem, which connects the brain with the spinal cord and is responsible for many vital functions such as breathing, maintaining the heart rate or swallowing. Secondly, a group of neurons deep within the brain called the basal ganglia, which has a role in voluntary movement.

But although all of a patient's neurons carry the mutation responsible for their Leigh syndrome, not every neuron is harmed by it. Knowing which neurons are affected, and why, could help develop treatments. Bolea, Gella, Sanz et al. therefore introduced the same Leigh syndrome mutation into different groups of neurons in three groups of mice. The first group had the mutation in the neurons that activate other cells, called glutamatergic or 'go' neurons. The second group had the mutation in the neurons that inhibit other cells, known as GABAergic, or 'stop', neurons. The third had the mutation in cholinergic neurons, which carry information from the brain to the organs.

Examining the mice revealed that having faulty mitochondria in GABAergic neurons from the basal ganglia and in glutamatergic neurons of the brainstem, but not in cholinergic neurons, leads to the symptoms of Leigh syndrome. The fault in the GABAergic neurons causes the epilepsy associated with the syndrome, while faulty mitochondria in the glutamatergic neurons give rise to the observed impairments in movement and breathing. This work could help researchers identify the cellular mechanisms that make neurons more or less resistant to the effects of faulty mitochondria. This in turn will provide a stepping stone to developing new treatments, which can then be tested on the mice developed for these experiments.

DOI: https://doi.org/10.7554/eLife.47163.002

specificity (*Arii and Tanabe, 2000*), a common feature shared by mitochondrial diseases, affecting high energy-requiring tissues such as muscle and brain (*Molnar and Kovacs, 2017*). Pathologically, LS is characterized by the presence of bilateral symmetrical lesions predominantly in the brainstem and basal ganglia (*Arii and Tanabe, 2000*). Neuronal damage is responsible for most of the fatal symptoms, including respiratory failure and seizures (*Barends et al., 2016*). However, the identity of the affected cellular populations and the molecular determinants of neuronal vulnerability have not been adequately elucidated, representing a challenge for the development of efficient treatments.

Mutations affecting the NDUFS4 subunit of mitochondrial Complex I, a key structural component for the assembly, stability and activity of the complex (*Calvaruso et al., 2012*), are commonly associated with a severe, early-onset LS phenotype (*Ortigoza-Escobar et al., 2016*). Although a late-onset case of LS has been recently reported (*Bris et al., 2017*), prognosis is usually poor and most of the patients die in early childhood (*Sofou et al., 2014*).

Animals with a global deletion of *Ndufs4* (Ndufs4KO mice) develop a fatal encephalomyopathy, which recapitulates the classical signs of LS, including motor alterations, respiratory deficits, epilepsy and premature death (*Quintana et al., 2010*; *Kruse et al., 2008*). Behavioral and neuropathological characterization of Ndufs4KO mice revealed the pivotal role of the dorsal brainstem, particularly the vestibular nucleus (VN), in disease manifestation and progression (*Quintana et al., 2012*). However, the genetic identity of the neuronal populations and circuitries involved in the plethora of symptoms observed have not yet been identified.

Here, we describe the contribution of genetically defined, discrete neuronal populations to the fatal phenotype of Ndufs4KO mice. To that end, we generated three mouse lines using a conditional genetic approach that selectively inactivates *Ndufs4* in glutamatergic (Vglut2-expressing),

GABAergic (*Gad2*-expressing) or cholinergic (ChAT-expressing) neurons. The results reveal distinct, lethal phenotypes for the glutamatergic and GABAergic neuronal populations.

## Results

### Reduced lifespan and body weight in cell type-specific conditional Ndufs4KO mice

To dissect the neuronal cell types contributing to the neuropathology observed in Ndufs4KO mice (*Quintana et al., 2010*; *Kruse et al., 2008*; *Quintana et al., 2012*), we generated three mouse lines lacking *Ndufs4* selectively in glutamatergic (expressing Vglut2), GABAergic or cholinergic neurons. We did this by crossing *Ndufs4* exon2-floxed mice with Cre-driver lines of mice expressing either *Slc17a6-Cre, Gad2-Cre* or *Chat-Cre* as described in *Figure 1A* and *Figure 1—figure supplement 1*; the affected mice are referred to here as: Vglut2:Ndufs4cKO, Gad2:Ndufs4cKO or ChAT:Ndufs4cKO mice and their respective controls are Vglut2:Ndufs4cCT, Gad2:Ndufs4cCT or ChAT:Ndufs4cCT.

*Ndusf4* gene inactivation in glutamatergic or GABAergic neurons of both male and female mice resulted in failure to thrive and premature death (*Figure 1B–F*); however, there was no effect on survival, body weight, or motor function when *Ndufs4* expression was abolished in cholinergic neurons (*Figure 1—figure supplement 1*). Selective deletion of *Ndufs4* in glutamatergic or GABAergic neurons was confirmed by western blot analysis of NDUFS4 levels in brain areas where Vglut2 or Gad2 are preferentially expressed (*Lein et al., 2007*) (*Figure 1—figure supplement 2*). Vglut2:Ndufs4cKO mice had a median lifespan of 67 days with a mortality rate of 90% at postnatal day (P) 128. Similarly, Gad2:Ndufs4cKO mice had a median lifespan of 60 days (90% mortality at P70) (*Figure 1B*). This premature death was preceded by a reduction in body weight gain in male and female mice of both genotypes (*Figure 1C–F*). At 7 weeks of age for females and 9 weeks of age for males, Vglut2:Ndufs4cKO mice stopped gaining weight, which resulted in an overall reduction in body weight when compared to age-matched controls (*Figure 1C,D*). Similarly, male and female Gad2:Ndufs4cKO mice body weight reached a plateau 2–3 weeks before manifesting a sudden unexpected death (*Figure 1E,F*). Both Vglut2:Ndufs4cKO and Gad2:Ndufs4cKO mice were also significantly smaller than their littermate controls (*Figure 1G–H*). This lack of weight gain and reduced size appeared to be due to decreased food intake in both genotypes (*Figure 1I,J*); however, this was not significantly different when food consumption was normalized to body weight (*Figure 1—figure supplement 2*).

### Vglut2:Ndufs4cKO and Gad2:Ndufs4cKO mice manifest clinically distinct phenotypes

The reduced lifespan and decreased body weight observed in both Vglut2:Ndufs4cKO and Gad2:Ndufs4cKO mice were the result of two prominently different clinical presentations. Gad2:Ndufs4cKO mice were, for the most part, phenotypically indistinguishable from controls, without any overt clinical alteration beyond a reduced growth rate for a few weeks prior to a sudden premature death. On the other hand, Vglut2:Ndufs4cKO mice manifested progressive motor and respiratory deterioration with most of the clinical signs visibly apparent (*Table 1*).

Vglut2:Ndufs4cKO mice presented body tremor and a decline in balance as early as at 5 weeks (early stage of the disease), which dramatically worsened as the disease progressed. In a mid-stage of the disease, mice increased body tremor and showed a prominent decline in balance and motor coordination (*Table 1*). Subsequently, animals started exhibiting ataxia and a progressive loss of the righting and hindlimb extension reflexes (clasping) (*Figure 1—figure supplement 3*). At a late stage of the disease, these mice showed increased tremor, were completely docile and hypotonic (*Table 1*). Animals also had difficulty maintaining a regular breathing pattern at an early stage of the disease. These breathing abnormalities worsened as the disease progressed with mice presenting noticeably shorter and deeper respirations at advanced stages of the disease (*Video 1*). Furthermore, at a late stage of the disease (over P60), about 40% of Vglut2:Ndufs4cKO developed hindlimb dragging and eventually hindlimb paralysis. We observed increased glial reactivity but no neuronal loss in the spinal cord of these mice when compared to Vglut2:Ndufs4cCT mice, which was determined by immunoblot analysis of spinal cord lysates from Vglut2:Ndufs4cCT and Vglut2:Ndufs4cKO mice using antibodies against GFAP (glial fibrillary acidic protein), Iba-1 (ionized calcium-binding

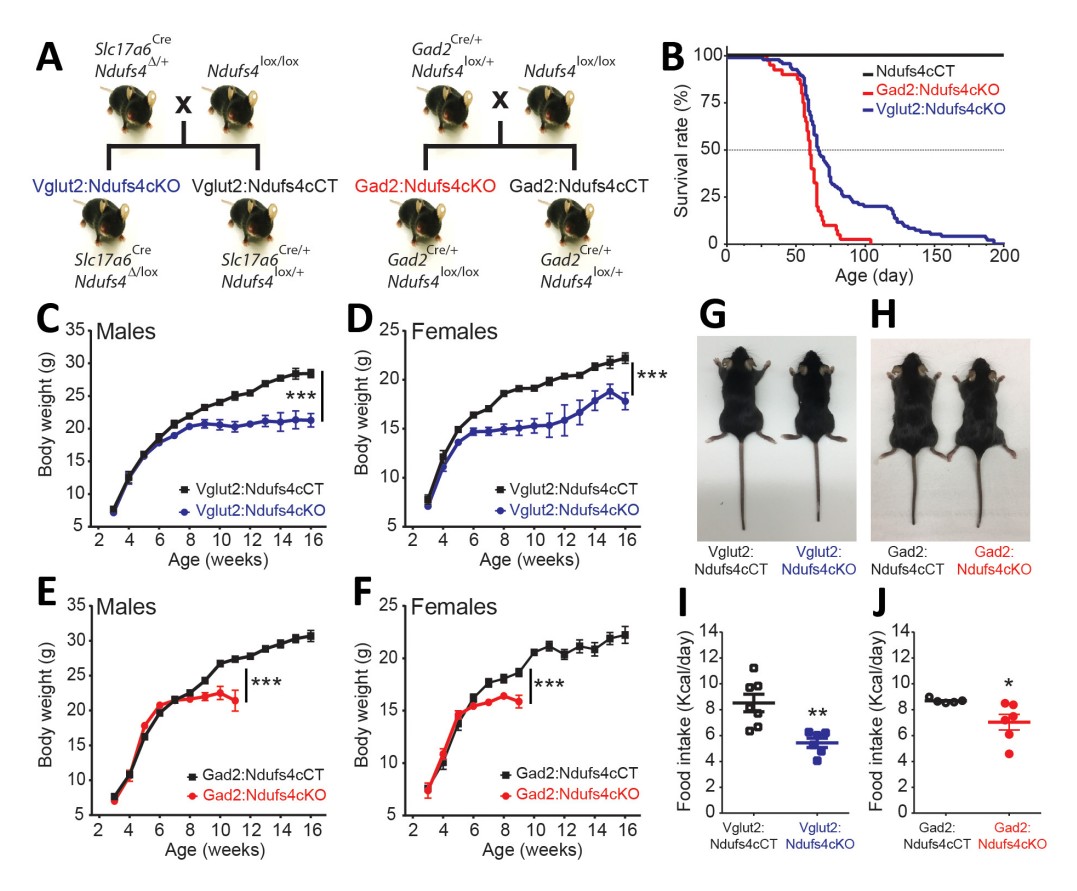

**Figure 1.** Generation of two mouse lines lacking *Ndufs4* selectively in Vglut2-expressing glutamatergic neurons or Gad2-expressing GABAergic neurons. (**A**) Breeding strategy to obtain mice with conditional *Ndufs4* deletion in glutamatergic neurons (Vglut2:Ndufs4cKO) or GABAergic neurons (Gad2:Ndufs4cKO) and their respective controls (Vglut2:Ndufs4cCT, Gad2:Ndufs4cCT). (**B**) Kaplan-Meier survival curve for Vglut2:Ndufs4cKO (n = 96; blue), Gad2:Ndufs4cKO (n = 40; red) and cCT mice (n = 50; black). (**C–D**) Body weight curves for Vglut2:Ndufs4cKO (n = 57 males, n = 48 females; blue) and Vglut2:Ndufs4cCT (n = 52 males, n = 50 females; black). (**E–F**) Body weight curves for Gad2:Ndufs4cKO (n = 30 males, n = 32 females; red) and Gad2:Ndufs4cCT (n = 41 males, n = 34 females; black). Data are presented as the mean ± SEM. Statistical analysis was performed using a two-way ANOVA (***p<0.001). (**G**) Representative images showing reduced body size in Vglut2:Ndufs4cKO mice when compared to Vglut2:Ndufs4cCT mice at P65. (**H**) Gad2:Ndufs4cKO mice also display reduced body size when compared to control littermates (Gad2:Ndufs4cCT) at P70. Food intake values (kcal/day) for (**I**) Vglut2:Ndufs4cKO (n = 6; closed blue squares) and Vglut2:Ndufs4cCT mice (n = 7; open black squares) and (**J**) Gad2:Ndufs4cKO (n = 6; closed red circles) and Gad2:Ndufs4cCT mice (n = 5; open black circles) at 8 weeks of age. Data are presented as the mean ± SEM. Statistical analysis was performed using an unpaired *t*-Test (**p<0.01, *p<0.05).

DOI: https://doi.org/10.7554/eLife.47163.003

The following figure supplements are available for figure 1:

**Figure supplement 1.** Generation and characterization of a conditional mouse line lacking *Ndufs4* in ChAT-expressing neurons.
DOI: https://doi.org/10.7554/eLife.47163.004

**Figure supplement 2.** NDUFS4 levels and food intake after conditional deletion of *Ndufs4* in Vglut2- and Gad2-expressing cells.
DOI: https://doi.org/10.7554/eLife.47163.005

**Figure supplement 3.** Reactive gliosis in the spinal cord of Vglut2:Ndufs4cKO mice.
DOI: https://doi.org/10.7554/eLife.47163.006

adapter molecule 1) and NSE (neuronal specific enolase) (*Figure 1—figure supplement 3*). This glial reactivity was further confirmed by immunofluorescence analysis using anti-GFAP and anti-Iba-1 antibodies in the spinal cords of the Vglut2:Ndufs4cKO mice that exhibited hindlimb dragging (*Figure 1—figure supplement 3*, top panels). However, no signs of demyelination or immune cell infiltration were observed (*Figure 1—figure supplement 3*, mid and bottom panels).

# Neuronal identity defines distinct neuroinflammatory patterns after *Ndufs4* deletion

LS is characterized by symmetrical brain lesions and neuroinflammation in select nuclei, predominantly brainstem and/or basal ganglia (*Arii and Tanabe, 2000*). Accordingly, late-stage *Ndufs4*-deficient mice present overt lesions in brainstem (namely vestibular nucleus (VN), cerebellar fastigial nucleus (FN) and inferior olive (IO)), olfactory bulb and basal ganglia (*Quintana et al., 2010*; *Quintana et al., 2012*; *Chen et al., 2017a*). To define the contribution of *Ndufs4* deficiency in either excitatory or inhibitory neurons to the overall neuroinflammatory phenotype and identify the specific brain areas with exacerbated astroglial and microglial reactivity, we performed immunofluorescence analysis using anti-GFAP and anti-Iba1 antibodies on brain sections of Vglut2:Ndufs4cKO and Gad2:Ndufs4cKO mice, and their respective controls (*Figure 2A–D*). Analysis of these sections showed marked glial reactivity in VN, IO and FN (*Figure 2A, C*; *Figure 2—figure supplement 1*), accompanied by increased caspase eight activation in affected areas (such as the VN) in Vglut2:Ndufs4cKO mice (*Figure 2E,F*), recapitulating most of the neuroinflammatory profile described for the global Ndufs4KO mice (*Quintana et al., 2010*). In contrast, Gad2:Ndufs4cKO mice presented a more restricted glial reactivity pattern, including marked microglial and astroglial reactivity in primarily GABAergic nuclei such as the external globus pallidus (GPe) in the basal ganglia and the substantia nigra pars reticulata (SNr) (*Figure 2B, D*), without affecting neighboring non-GABAergic areas like the dopaminergic substantia nigra pars compacta (SNc) (*Figure 2—figure supplement 2*). Other prominently GABAergic areas such as the olfactory bulb also showed increased immunoreactivity for GFAP and Iba-1 in Gad2:Ndufs4cKO mice when compared to control mice (*Figure 2B,D*; *Figure 2—figure supplement 1*). In a percentage of Gad2:Ndufs4cKO mice, prominent microglial reactivity with Purkinje neuron loss (as assessed by calbindin staining) was also observed in the cerebellar vermis and flocculus (*Figure 2—figure supplement 2*). Normal *Slc17a6* (Vglut2) or *Gad2* transcript abundance was observed in the VN and OB of late-stage Vglut2:Ndufs4cKO or Gad2:Ndufs4cKO mice, respectively, suggesting preservation of *Ndufs4*-deficient neuronal populations (*Figure 2G,H*).

The extensive neuroinflammation present in the brainstem of Vglut2:Ndufs4cKO mice, and its similarity to the phenotype of the global Ndufs4KO, allowed us to further define this inflammatory phenotype using whole-tissue transcriptional profiling. Gene expression analysis in the brainstem of late-stage (over P68) Vglut2:Ndufs4cKO mice using Illumina Beadchips (MouseRef-8 V2; Illumina) showed that differentially expressed (DE) mRNAs were, for the most part, upregulated in Vglut2:Ndufs4cKO mice (p<0.05, Fold Change > 2) (*Figure 2—figure supplement 2*). Selected genes included transcripts directly involved in the regulation of the immune system such as chemokines and their receptors (*Ccl3*, *Ccl4* and *Cxcr3*), toll-like receptors (*Tlr2*, *Tlr7*), complement proteins (*C1qa*, *C1qb*, *C3* and *C4b*), surface antigens (*Cd84*, *Cd86* and *Ly9*), and markers of the myeloid cell lineage (infiltrating macrophages and microglia) including *Lyz*, *Lyz2*, *Lyzs* and *Slc11a1*, among others.

**Table 1.** Neurological signs observed in Vglut2:Ndufs4cKO mice according to the stage of disease.

|  | Early | Mid | Late |
|---|---|---|---|
| *Body tremor* | + | ++ | +++ |
| *Motor alterations* |  |  |  |
| *Decline in balance* | + | ++ | +++ |
| *Ataxia* | - | +/++ | +++ |
| *Loss of motor coordination* | - | +/++ | +++ |
| *Inability of righting* | - | + | ++/+++ |
| *Hindlimb clasping* | - | + | ++ |
| *Respiratory abnormalities* |  |  |  |
| *Breathing irregularities* | + | ++ | +++ |

-, absent; +, mild; ++, moderate; +++, severe.
DOI: https://doi.org/10.7554/eLife.47163.007

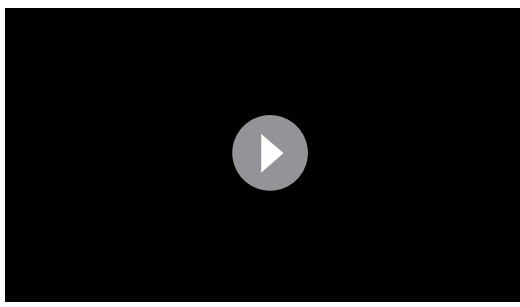

**Video 1.** Breathing irregularities in a late-stage Vglut2:
Ndufs4cKO mice.
DOI: https://doi.org/10.7554/eLife.47163.008

Functional enrichment analysis of differentially expressed mRNAs (1.4-fold or higher) using over-representation analysis (ORA) (*Wang et al., 2017*) showed that the top 10 most overrepresented Gene Ontology (GO) categories (biological process, non-redundant) were all related to defense and immune responses, and also uncovered components of the adaptive immune response (GO:0002250; p=2.0177e-12, FDR = 7.2033e-10) including 'leucocyte mediated immunity' (GO:0002443; p=1.7533e-12, FDR = 7.2033e-10) and 'myeloid leukocyte activation' (GO:0002274; p=8.7458e-10, FDR = 9.0537e-8) (*Figure 2—figure supplement 2*). Therefore, to further characterize the immune cell composition in these whole tissue gene expression profiles, we applied recently developed deconvolution tools that use leucocyte gene expression signature matrices to computationally infer the relative proportions of each immune cell type in gene expression mixtures (*Newman et al., 2015*; *Chen et al., 2017b*) (*Figure 2—figure supplement 2*). This analysis revealed a gene expression profile consistent with an increased proportion of proinflammatory CD4 cells (Follicular cells, Th1, Th17 and Treg), dendritic cells, mast cells and macrophages, and an underrepresentation of CD4 Th2 cells, CD8 cells and NK cells in the brainstem of Vglut2:Ndufs4cKO mice compared to controls, showing that *Ndufs4* deficiency in glutamatergic neurons promotes a neuroinflammatory environment that involves a commitment to distinct proinflammatory $T_H$ cell lineages and a defined profile of tissue defense cells.

## Vglut2:Ndufs4cKO but not Gad2:Ndufs4cKO mice present motor alterations and respiratory deficits

Motor dysfunction is a prominent feature in LS and Ndufs4KO mice pathology (*Sofou et al., 2014*; *Quintana et al., 2010*). To genetically define the neuronal cell types mediating this functional disorder, we assessed motor coordination in Vglut2:Ndufs4cKO mice, Gad2:Ndufs4cKO mice, and their respective controls (*Figure 3*). Starting at P40, and concurring with the onset of clinical signs, Vglut2:Ndufs4cKO mice showed impaired rotarod performance when compared to control littermates (*Figure 3A*). While control mice maintained rotarod performance, Vglut2:Ndufs4cKO mice failed to properly execute the task, presenting a progressive decline in motor coordination, in line with the clinical phenotype. Conversely, and in agreement with the lack of apparent clinical signs, no differences in rotarod performance were observed in Gad2:Ndufs4cKO mice compared to control littermates (*Figure 3B*). Exposure to a novel environment revealed a hypoactive phenotype in Vglut2:Ndufs4cKO mice as assessed by a reduction in the total distance traveled and the speed of exploratory movement in the open-field test (*Figure 3C–E*). The severity of this phenotype in Vglut2:Ndufs4cKO mice positively correlated with age and disease stage (*Figure 3—figure supplement 1*). In contrast, no significant differences were observed in either distance traveled or speed in the open-field test between Gad2:Ndufs4cKO mice and their respective controls (*Figure 3F–H*). No differences in the innervation of neuromuscular junctions were observed in the gastrocnemius muscle of either Vglut2:Ndufs4cKO or Gad2:Ndufs4cKO mice (*Figure 3I–K*), suggesting a central origin of the motor phenotype.

Respiratory abnormalities are frequently associated with disease mortality in LS patients and global Ndufs4KO mice (*Arii and Tanabe, 2000*; *Quintana et al., 2012*; *Gerards et al., 2016*). To provide insight into the genetic identity of the neurons responsible for this respiratory phenotype, we assessed respiratory function by unrestrained whole-body plethysmography in awake Vglut2:Ndufs4cKO and Gad2:Ndufs4cKO mice. Vglut2:Ndufs4cKO mice exhibited erratic plethysmographic recordings (*Figure 4A*). In these mice, the frequency of respiration ($f_R$) was markedly reduced at a mid-stage (P40-P60) of the disease and worsened as disease progressed (*Figure 4C*). In addition, significant differences were also seen in the volume of air inspired by the animal during one breath (tidal volume, $V_T$). $V_T$ was increased in Vglut2:Ndufs4cKO mice at a mid-stage of the disease and became significantly larger than in Vglut2:Ndufs4cCT at late disease stages (*Figure 4D*). In contrast,

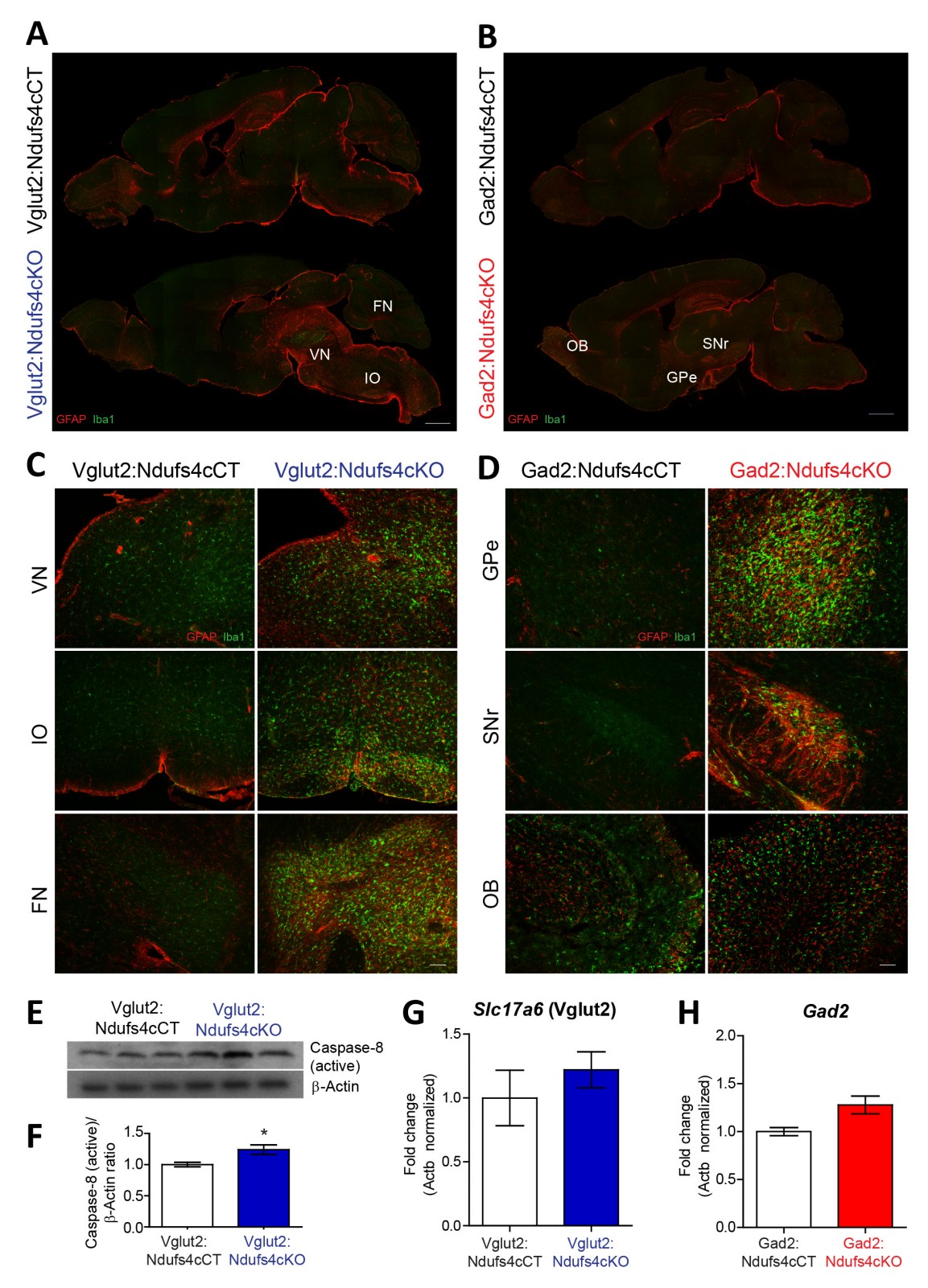

**Figure 2.** Distinct histopathological pattern in Vglut2:Ndufs4cKO and Gad2:Ndufs4cKO mice. (**A**) Whole-mount composite sagittal sections showing GFAP (red) and Iba-1 (green) staining in late-stage Vglut2:Ndufs4cKO mice and their respective controls (Vglut2:Ndufs4cCT). Affected areas are indicated: Vestibular nucleus (VN), inferior olive (IO) and fastigial nucleus (FN). (**B**) Whole-mount composite sagittal sections showing GFAP (red) and Iba-1 (green) staining in Gad2:Ndufs4cKO mice and their respective controls (Gad2:Ndufs4cCT). Affected areas are indicated: External globus pallidus

*Figure 2 continued on next page*

*Figure 2 continued*

(GPe), the sustantia nigra pars reticulata (SNr) and the olfactory bulb (OB). (C) Close-up micrographs showing GFAP and Iba1 staining in the VN, IO, and FN of late-stage Vglut2:Ndufs4cKO and Vglut2:Ndufs4cCT mice. (D) Close-up micrographs showing GFAP and Iba-1 staining in the GPe, SNr and OB in Gad2:Ndufs4cKO and Gad2:Ndufs4cCT mice. Scale bar A-B:1000 µm. C-D: 50 µm. (E–F) Western blot analysis (E) and quantification (F) for active caspase eight and β-Actin (loading control) levels in the vestibular nucleus of late-stage Vglut2:Ndufs4cKO mice (n = 3) and Vglut2:Ndufs4cCT mice (n = 3). (G) *Slc17a6* (Vglut2) transcript levels in the vestibular nucleus of late-stage Vglut2:Ndufs4cKO mice (n = 3) and Vglut2:Ndufs4cCT mice (n = 3). (H) *Gad2* transcript levels in the olfactory bulb of Gad2:Ndufs4cKO mice (n = 3) and Gad2:Ndufs4cCT mice (n = 3). Data are presented as the mean ± SEM. Statistical analysis was performed using an unpaired *t*-Test (*p<0.05).

DOI: https://doi.org/10.7554/eLife.47163.010

The following figure supplements are available for figure 2:

**Figure supplement 1.** Histopathological pattern in Vglut2:Ndufs4cKO and Gad2:Ndufs4cKO mice.

DOI: https://doi.org/10.7554/eLife.47163.011

**Figure supplement 2.** *Ndufs4* deficiency in *Gad2*-expressing GABAergic and Vglut2-expressing glutamatergic cells results in prominent neuroinflammation in specific areas.

DOI: https://doi.org/10.7554/eLife.47163.009

Gad2:Ndufs4cKO mice did not exhibit irregular plethysmographic traces and had breathing patterns ($f_R$ and $V_T$) that did not differ from controls (*Figure 4B and E–F*). These data reveal that the motor impairment and respiratory deficits observed in the Ndufs4KO mouse are mediated by Vglut2-expressing excitatory neuronal populations.

We have shown that neuronal inactivation of *Ndufs4* in the VN promotes breathing abnormalities (*Quintana et al., 2012*). This observation, along with the extensive neuroinflammation observed in the VN of late-stage Vglut2:Ndufs4cKO mice, prompted us to assess the activity of Vglut2-expressing neurons in the VN using in vivo electrophysiology. Cell-firing activity was recorded during an open-field session before identification by laser stimulation of Channelrhodopsin-2 (ChR2)-expressing vestibular glutamatergic neurons in Vglut2:Ndufs4cKO and Vglut2:Ndufs4cCT mice (*Figure 5A*). Electrophysiological recordings in vivo showed that the firing rate for vestibular Vglut2-expressing neurons from Vglut2:Ndufs4cKO mice was not significantly different than controls when mice were at rest. However, when the mice were actively moving in the open-field, the Vglut2-expressing VN neurons in the control mice approximately doubled their firing rate to about 40 Hz but the experimental group did not (*Figure 5B*). The failure of VN glutamatergic neurons in Vglut2:Ndufs4cKO mice to respond to motor activity may account for the breathing abnormalities.

## Abnormal body temperature regulation in Gad2:Ndufs4cKO mice

Dysregulation of body temperature homeostasis, such as hypothermia, is commonly observed in LS patients (*Finsterer, 2008*). Accordingly, global Ndufs4KO mice present reduced body temperature starting at P30 (*Quintana et al., 2010*). However, the genetic identity of the neuronal population involved in this phenotype was unknown. Hence, using telemetric temperature monitoring, we observed that Vglut2:Ndufs4cKO presented normal resting body temperature during early life, decreasing only as the disease progressed to mid-late stage (P40 onwards, *Figure 6A*). In contrast, we identified a severe reduction in resting body temperature of Gad2:Ndufs4cKO as early as P20-P30, which was maintained at all time points analyzed (*Figure 6B*). Therefore, our results suggest a central role of GABAergic neurons in central body temperature regulation in the context of *Ndufs4* deficiency.

## Gad2:Ndufs4cKO manifest sudden unexpected death associated with epilepsy

In Gad2:Ndufs4cKO mice, spontaneous seizure-like events were observed during routine husbandry practices such as lifting the animal by the tail or cage cleaning (*Video 2*). To define whether fatal seizures were the cause of the unanticipated death observed in Gad2:Ndufs4cKO mice, starting at P40 animals were continuously video-recorded to monitor for potential seizures leading to an abrupt death. Analysis of the recordings revealed that all deaths in Gad2:Ndufs4cKO mice consistently followed a severe generalized tonic-clonic convulsion. In contrast, spontaneous, sporadic or lethal seizures were not observed in either controls (data not shown) or Vglut2:Ndufs4cKO mice at any stage of the disease (*Figure 7—figure supplement 1*). Detailed visualization and analysis of the images

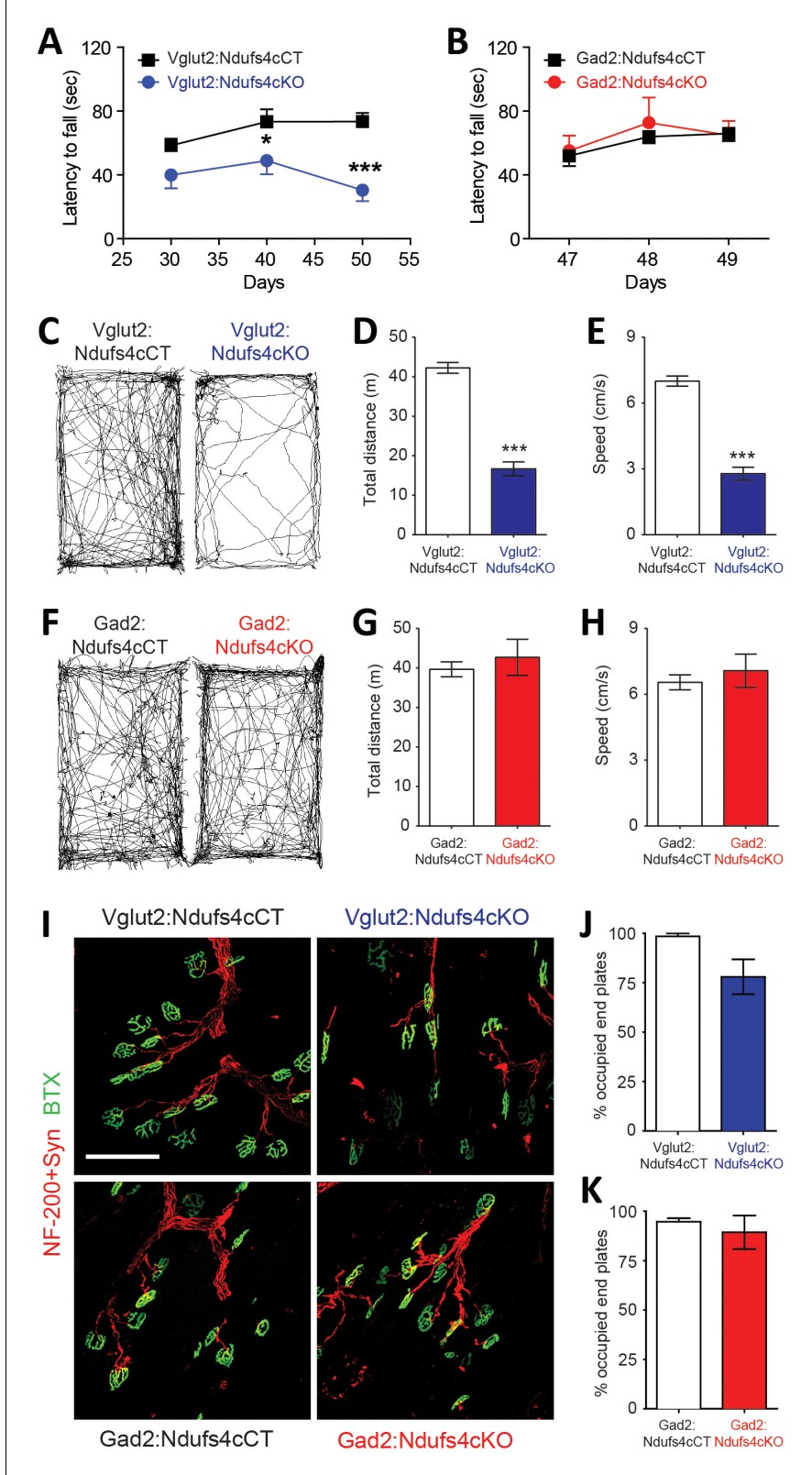

**Figure 3.** Motor impairment in Vglut2:Ndufs4cKO mice. Latency to fall (seconds) in the Rotarod test for (**A**) Vglut2:Ndufs4cKO (n = 8) and Vglut2:Ndufs4cCT mice (n = 10), and (**B**) Gad2:Ndufs4cKO (n = 4) and Gad2:Ndufs4cCT mice (n = 10). Statistical analysis was performed using two-way ANOVA followed by Bonferroni post-test (*p<0.05, ***p<0.001). Representative mouse track plots, total distance traveled and average speed during 10 min in the

*Figure 3 continued on next page*

*Figure 3 continued*

open-field for (C–E) Vglut2:Ndufs4cKO (n = 36) and Vglut2:Ndufs4cCT mice (n = 70), and (F–H) Gad2:Ndufs4cKO (n = 11) and Gad2:Ndufs4cCT mice (n = 16). Statistical analysis was performed using an unpaired *t*-Test (***p<0.001). (I) Representative images of gastrocnemius neuromuscular junctions of Vglut2:Ndufs4cKO mice and Gad2:Ndufs4cKO mice and their respective controls (scale bar: 100 µm). (J) Proportion of fully innervated neuromuscular junctions in the gastrocnemius muscle of late-stage Vglut2:Ndufs4cKO and Vglut2:Ndufs4cCT mice (n = 3 mice per group). (K) Quantification of fully innervated neuromuscular junctions in the gastrocnemius muscle of the Gad2:Ndufs4cKO and Gad2:Ndufs4cCT mice (n = 3 mice per group). Statistical analysis was performed using a Mann-Whitney test. Data are presented as the mean ± SEM.
DOI: https://doi.org/10.7554/eLife.47163.012

The following figure supplement is available for figure 3:

**Figure supplement 1.** Age-dependent motor decline in Vglut2:Ndufs4cKO mice.
DOI: https://doi.org/10.7554/eLife.47163.013

---

unveiled that seizures in Gad2:Ndufs4cKO mice were mostly generalized and of multiple semiology (unilateral, primary bilateral, secondary bilateral or alternating). Subsequent electroencephalographic (EEG) and electromyographic (EMG) characterization showed the presence of spontaneous generalized tonic-clonic (GTC) seizures (*Figure 7A*), interictal spikes and myoclonic seizures in Gad2: Ndufs4cKO mice (*Figure 7B–C*), which are hallmarks of epilepsy in mitochondrial disorders (*Rahman, 2015*), with 40–60% of mitochondrial disease patients manifesting seizures (*Koenig, 2008*; *El Sabbagh et al., 2010*; *Canafoglia et al., 2001*). In addition, local field potential (LFP) recordings identified the presence of seizure-like events (*Queiroz et al., 2009*) in the GPe of Gad2:Ndufs4cKO mice as soon as P35, while being absent in control mice (*Figure 7D*),

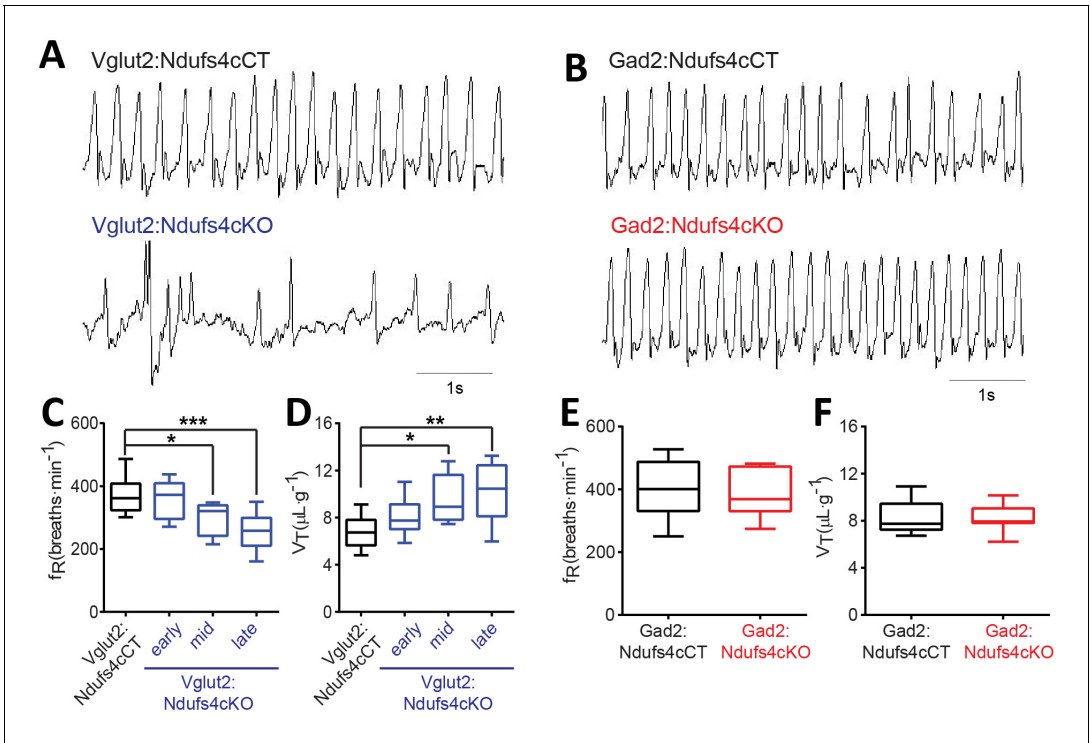

**Figure 4.** Breathing alterations in Vglut2:Ndufs4cKO mice. Representative 5 s plethysmographic recordings from (A) Vglut2:Ndufs4cCT mice (top), late-stage Vglut2:Ndufs4cKO mice (bottom), and (B) Gad2:Ndufs4cCT mice (top) and Gad2:Ndufs4cKO mice (bottom). Respiratory frequency ($f_R$, breaths·min$^{-1}$) and tidal volume ($V_T$, µL·g$^{-1}$) for (C–D) Vglut2:Ndufs4cCT (n = 12) and Vglut2:Ndufs4cKO mice at different stages of the disease (early, n = 7; mid, n = 8; late, n = 12) and (E–F) Gad2:Ndufs4cCT (n = 7) and Gad2:Ndufs4cKO mice (n = 7). Data are presented as the mean ± SEM. Statistical analysis was performed using (C–D) one-way ANOVA followed by Bonferroni post-test (*p<0.05, **p<0.01, ***p<0.001) and (E–F) an unpaired *t*-Test.
DOI: https://doi.org/10.7554/eLife.47163.014

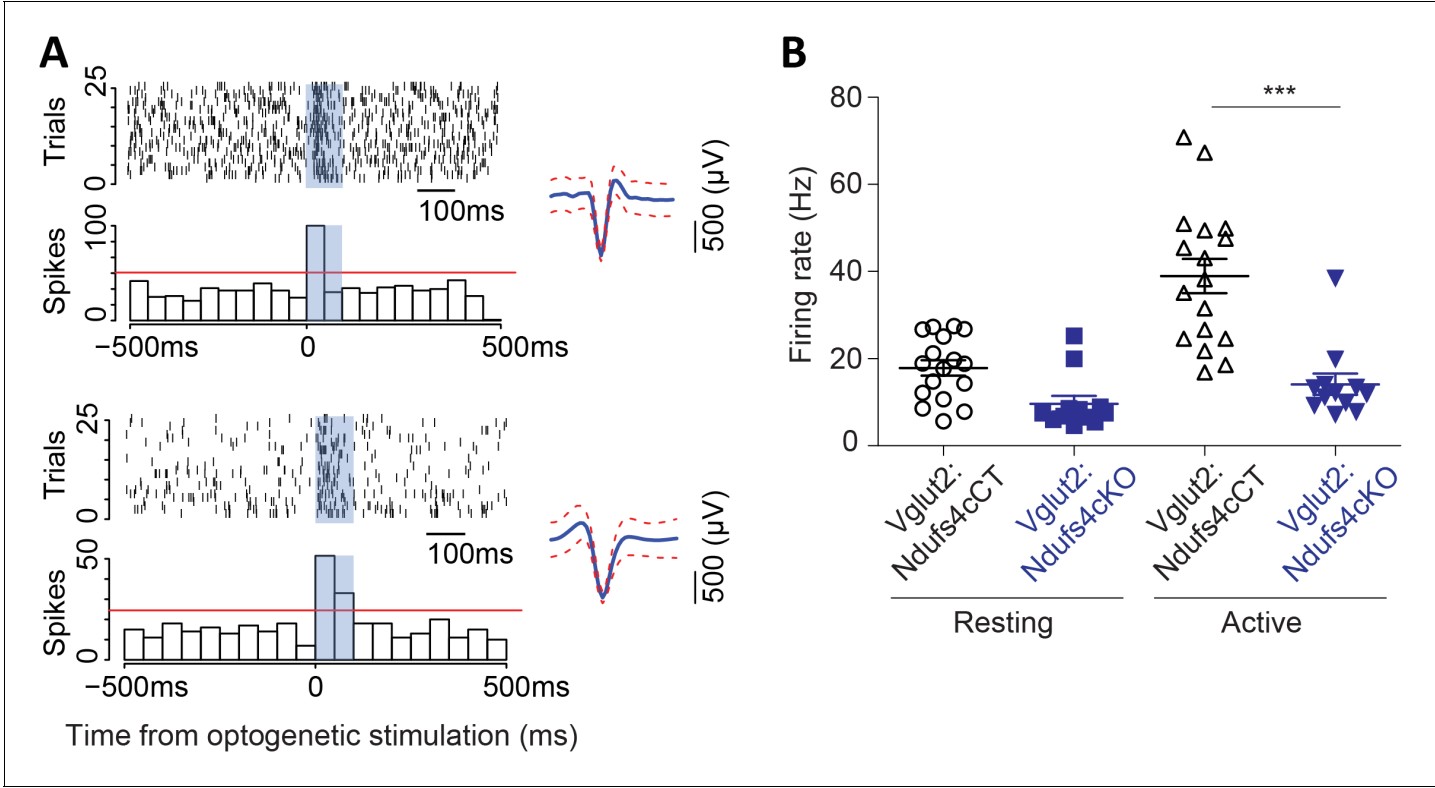

**Figure 5.** Vglut2-expressing glutamatergic cells in the vestibular nucleus show reduced in vivo electrophysiological activity in Vglut2:Ndufs4cKO mice. (A) Representative identification of two different glutamatergic cells with raster plots (upper part) and peri-stimulus histograms (PSTH, lower part) using optogenetic stimulation. 0 represents the onset of optogenetic stimulation. Red line represents mean firing rate from the 500 ms before the stimulus plus three time the SD. Blue shading indicates period of laser stimulation. (B) Firing rate of Vglut2-expressing glutamatergic cells (n = 17 from Vglut2: Ndufs4cCT mice and n = 12 from Vglut2:Ndufs4cKO mice) during a 10 min session in an open-field. Reduced electrophysiological activity was observed in freely-moving late-stage Vglut2:Ndufs4cKO mice when compared to Vglut2:Ndufs4cCT mice regardless behavioral state (resting or active). Data are presented as the mean ± SEM. Statistical analysis was performed using two-way repeated measures ANOVA with Bonferroni post-test (***p<0.001). DOI: https://doi.org/10.7554/eLife.47163.015

suggesting that subcortical alterations in affected regions may precede the development of generalized seizures.

In individuals with mitochondrial disease, seizures are commonly observed after stressors such as febrile events (*Bindoff and Engelsen, 2012*). Hence, susceptibility of Gad2:Ndufs4cKO mice to thermally-induced seizures was assessed (*Figure 7E–F*). Gad2:Ndufs4cKO mice were highly sensitive to temperature compared to Gad2:Ndufs4cCT mice, with some mice displaying seizures already at basal temperature. The number of seizures significantly increased with temperature; 50% of the mice exhibited seizures at 39.5°C and all Gad2:Ndufs4cKO manifested seizures by 41.5°C. Both spontaneous and temperature-induced seizures were always preceded by a myoclonic (MC) seizure (*Figure 7A and F*) with the latter being more severe (Racine scale stage 5) than the former (Racine scale stage 4) (data not shown). Both spontaneous and thermally-evoked seizures were characterized by EEG hyperactivity. Similarly to controls, Vglut2:Ndufs4cKO did not show susceptibility to temperature-induced seizures up to 42°C (data not shown). Peri-onset administration (from P40 onwards) of the anti-epileptic drugs levetiracetam (60 mg/kg), perampanel (0.75 mg/kg) or carbamazepine (40 mg/kg) to Gad2:Ndufs4cKO mice failed to prevent fatal epileptic events, resulting in similar lifespan (*Figure 7—figure supplement 2*). Drug-treated mice showed a median survival of 63, 62 and 66 days, respectively, which was not statistically significant when compared to the median lifespan of vehicle-treated mice (60 days).

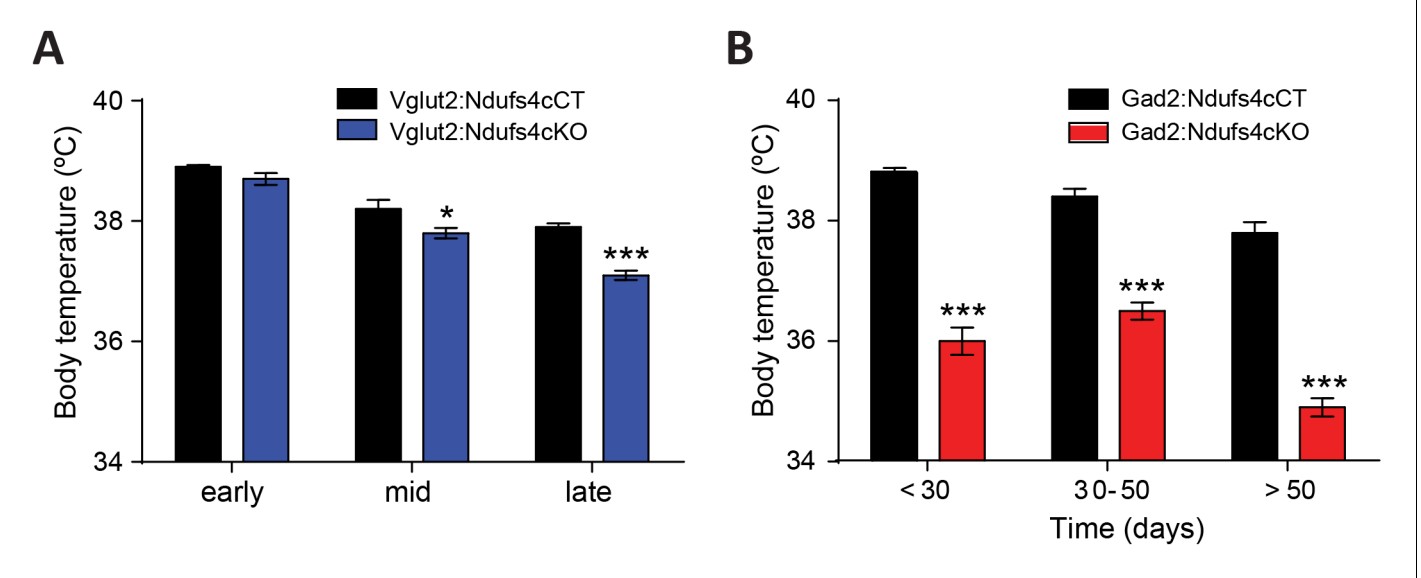

**Figure 6.** Decreased body temperature in Vglut2:Ndufs4cKO and Gad2:Ndufs4cKO mice. (**A**) Telemetric body temperature measurements in Vglut2:Ndufs4cKO (n = 4–8) and Vglut2:Ndufs4cCT mice (n = 7–10) at different stages of the disease. (**B**) Telemetric body temperature measurements in Gad2:Ndufs4cKO (n = 7–11) and Gad2:Ndufs4cCT mice (n = 5–7) at different ages. Data are presented as the mean ± SEM. Statistical analysis was performed using two-way ANOVA followed by Bonferroni post-test (*p<0.05, ***p<0.001).

DOI: https://doi.org/10.7554/eLife.47163.016

## Discussion

Neuronal vulnerability is one of the hallmarks of Leigh Syndrome. However, the restricted anatomical distribution of brain lesions indicates a clear gradation in neuronal susceptibility to LS-causing mutations. Specific neuronal populations in affected regions, such as brainstem or basal ganglia, are highly vulnerable to mitochondrial dysfunction and may underlie the plethora of neurological signs observed in LS. The scarcity and high variability of patients has limited our knowledge on the genetic identity and relative contribution of the affected neuronal populations to the phenotype; thus, a model system with consistent neuropathological features resembling mitochondrial disease is a valuable research tool. We have shown (*Quintana et al., 2010*; *Quintana et al., 2012*) that mice lacking the *Ndufs4* gene (Ndufs4KO mice) recapitulate the clinical signs of the human disease (*Ortigoza-Escobar et al., 2016*; *Quintana et al., 2012*). Ndufs4KO mice present overt lesions predominantly in the brainstem (*Quintana et al., 2010*; *Quintana et al., 2012*; *Johnson et al., 2013*), but also in the striatum in late stages of the disease (*Quintana et al., 2012*). Hence, we hypothesized that a concerted role of diverse neuronal populations was necessary to drive the plethora of symptoms observed in Ndufs4KO mice.

In this study, we use a conditional genetic approach to selectively ablate NDUFS4 in ChAT-expressing cholinergic neurons (ChAT:Ndufs4cKO), Vglut2-expressing glutamatergic neurons (Vglut2:Ndufs4cKO) or Gad2-expressing GABAergic neurons (Gad2:Ndufs4cKO), thus restricting Complex I deficiency to some of the most abundant neuronal populations. This approach allowed us to provide a comprehensive dissection of the neuronal involvement in the phenotype of LS.

Animals lacking *Ndufs4* in cholinergic neurons (ChAT:Ndufs4cKO) were phenotypically

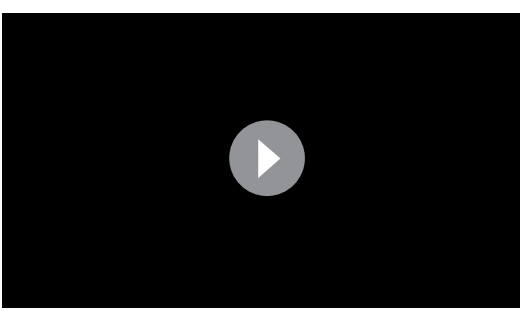

**Video 2.** Generalized tonic-clonic seizure without hypermotor (Racine scale stage 4) in Gad2:Ndufs4cKO mice.
DOI: https://doi.org/10.7554/eLife.47163.020

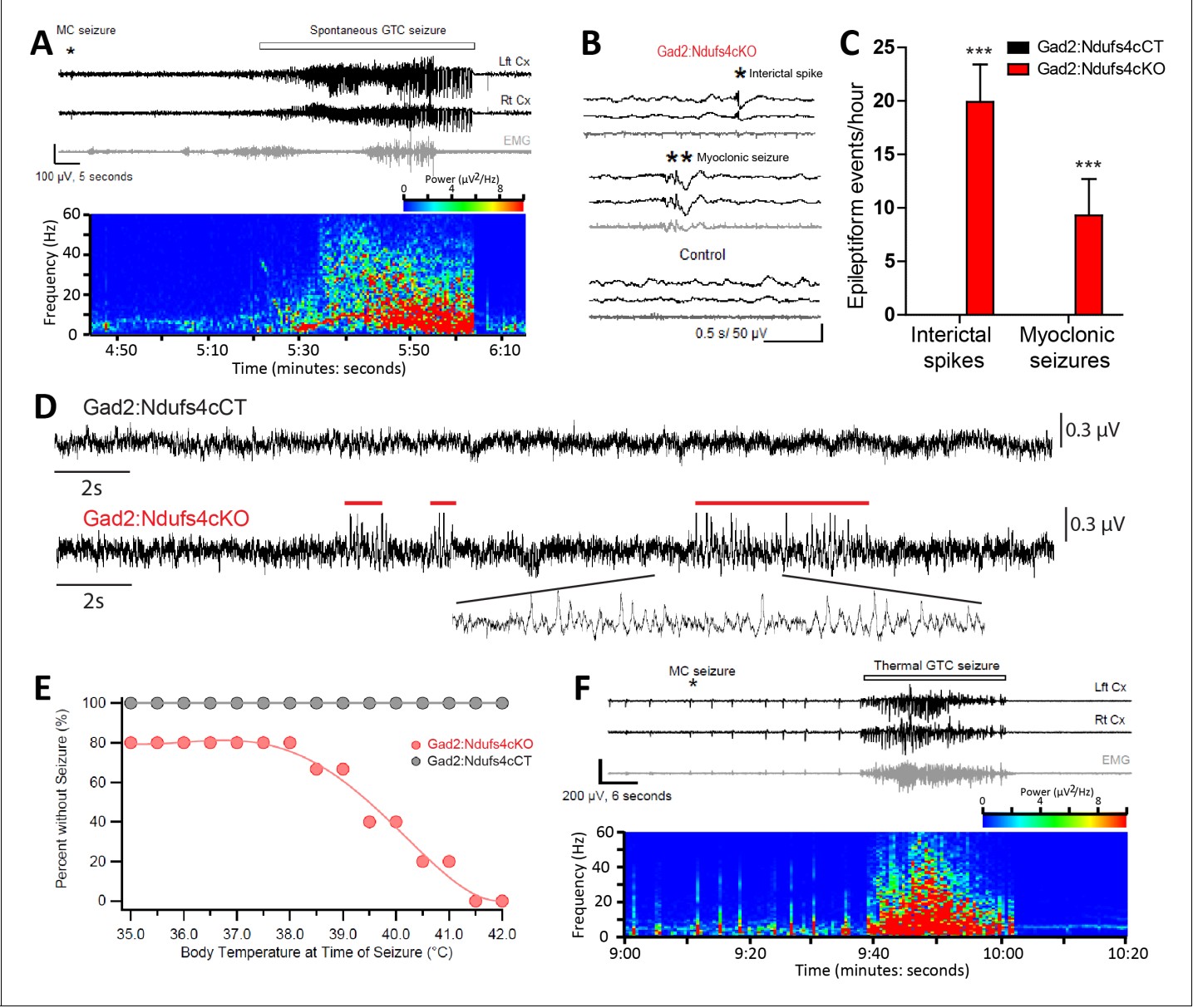

**Figure 7.** Gad2:Ndufs4cKO mice present epileptic seizures. (**A**) EEG-EMG recordings and spectrogram analysis showing the frequency and power density of a typical spontaneous seizure in a Gad2:Ndufs4cKO mouse. A preceding myoclonic seizure (MC) is marked with an asterisk. (**B**) Representative EEG-EMG traces and (**C**) number of epileptiform events (interictal spikes and myoclonic seizures) in Gad2:Ndufs4cKO (n = 9) and control (Gad2:Ndufs4cCT; n = 5) mice. Data are presented as the mean ± SEM. Statistical analysis was performed using two-way ANOVA (***p<0.001). (**D**) Local field potential (LFP) recordings in the globus pallidus of Gad2:Ndufs4cCT and Gad2:Ndufs4cKO mice at P35. Notice the presence of epileptic events (red lines) in the traces from Gad2:Ndufs4cKO mice. (**E**) Percentage of Gad2:Ndufs4cKO and Gad2:Ndufs4cCT mice remaining seizure-free after increasing body temperature. (**F**) Representative EEG-EMG recordings and spectrogram analysis of a thermally-induced seizure in a Gad2:Ndufs4cKO mouse. A preceding myoclonic seizure (MC) is marked with an asterisk. Lft Cx, left cortex; Rt Cx: right cortex; EMG: electromyography.
DOI: https://doi.org/10.7554/eLife.47163.017

The following figure supplements are available for figure 7:

**Figure supplement 1.** Vglut2:Ndufs4cKO mice do not present spontaneous seizures.
DOI: https://doi.org/10.7554/eLife.47163.018
**Figure supplement 2.** Lifespan of Gad2:Ndufs4cKO mice is not extended by antiepileptic treatment.
DOI: https://doi.org/10.7554/eLife.47163.019

**Table 2.** Clinical signs inNdufs4-LS patients, Ndufs4KO mice and conditional Vglut2:Ndufs4cKO and Gad2:Ndufs4cKO mice.

| | *Ndufs4*-LS patients | Ndufs4KO mouse | Vglut2:Ndufs4cKO mouse[*] | Gad2:Ndufs4cKO mouse[*] |
|---|---|---|---|---|
| *Decreased lifespan* | 22/22 | Yes | Yes | Yes |
| *Feeding impairment* | 8/22 | n.d | Yes | Yes |
| *Reduced body weight* | n.a. | Yes | Yes | Yes |
| *Brainstem lesions* | 14/22 | Yes | Yes | No |
| *Basal ganglia lesions* | 9/22 | Yes | No | Yes |
| *Ataxia, motor alterations* | 22/22 | Yes | Yes | No |
| *Growth retardation* | 11/22 | Yes | Yes | Yes |
| *Hypotonia* | 22/22 | Yes | Yes | No |
| *Respiratory abnormalities* | 22/22 | Yes | Yes | No |
| *Increased sensitivity to VAs* | n.a. | Yes | Yes[†] | No[†] |
| *Seizures* | 4/22 | Yes | No | Yes |
| *Hypothermia* | n.a. | Yes | Yes[‡] | Yes |

n.d, not determined; n.a, not available. [*]Reported in this study. [†]Reported in *Zimin et al. (2016)*. [‡]Only in late-stage mice. VAs: volatile anesthetics. Adapted from *Quintana et al. (2012)*.

DOI: https://doi.org/10.7554/eLife.47163.021

equivalent to controls, indicating no overt contribution of this cell type to the pathology observed in Ndufs4KO mice. In contrast, Vglut2:Ndufs4cKO and Gad2:Ndufs4cKO mice had reduced lifespan and body weight, which was accompanied by a decrease in food intake, which are common clinical signs that appear in Leigh Syndrome patients (*Rahman et al., 1996*; *Smeitink, 2003*). Recent reports have shown that neuronal cell-specific NDUFS4 knock down in *Drosophila* also leads to severe feeding abnormalities and premature death (*Foriel et al., 2018*). Our results indicate a conserved role for neurons in the onset and progression of the pathological condition of global *Ndufs4* deficiency and reveal that both glutamatergic and GABAergic systems contribute to the growth and lethality phenotype. Noteworthy, the slight reduction in phenotype severity of conditional neuronal *Ndufs4* deficiency, with relative neuronal preservation, compared to the global Ndufs4KO mice suggests a concerted contribution of different cell types. In this regard, our previous work has shown the importance of neuron-astrocyte crosstalk in the development of neurodegeneration in the context of mitochondrial disease (*Liu et al., 2015*). However, astrocytic *Ndufs4* deficiency is not sufficient to recapitulate the phenotype of global Ndufs4KO mice (*Ramadasan-Nair et al., 2019*), underscoring the central role of neurons in the disease. Even so, recent studies have shown enhanced neuronal survival in global Ndufs4KO after disruption of hepatic S6K1 (*Ito et al., 2017*). Hence, the role of cell and tissue non-autonomous effects on disease progression have to be taken into account to fully understand the phenotype of neuronal-specific conditional Ndufs4KO mice.

Apart from the premature death and feeding deficits, Vglut2:Ndufs4cKO and Gad2:Ndufs4cKO mice present two markedly distinct clinical entities (summarized in *Table 2*). With Vglut2:Ndufs4cKO mice, the lethality was associated with severe motor and respiratory alterations, whereas with Gad2: Ndufs4cKO mice, sudden unexpected death was associated with epilepsy (SUDEP) (*Abdel-Mannan et al., 2019*; *Manolis et al., 2019*; *DeGiorgio et al., 2019*) with no overt clinical alteration beyond the weight loss.

Histologically, Vglut2:Ndufs4cKO mice present with prominent neuroinflammation and lesions in areas of the brainstem such as the VN, IO and the cerebellar FN, reminiscent of the pathology found in Ndufs4KO mice (*Quintana et al., 2010*). We have identified a critical role of brainstem lesions in the development of fatal breathing alterations observed in the Ndufs4KO mice (*Quintana et al., 2012*), in agreement with human LS patients (*Arii and Tanabe, 2000*). Glutamatergic signaling in the brainstem has been shown to regulate breathing (*Whitney et al., 2000*). In addition, VN glutamatergic neurons have been suggested to modulate respiratory responses (*Xu et al., 2002*). Furthermore,

the Pre-Bötzinger (PreBotC) complex, a key respiratory center, is composed of glutamatergic neurons (*Stornetta et al., 2003*) and receives extensive glutamatergic inputs (*Bochorishvili et al., 2012*). We have shown that Ndufs4KO mice present intrinsic PreBotC alterations and that vestibular projections to the PreBotC are necessary for maintaining respiration (*Quintana et al., 2012*). Our electrophysiological recordings in Vglut2-expressing VN neurons from Vglut2:Ndufs4cKO mice show relatively normal basal firing rate but fail to increase spiking in response to locomotor activity. Neuronal firing is a highly energy-requiring process, mostly dependent in mitochondrial function (*Harris et al., 2012*), especially in glutamatergic synapses (*Juge et al., 2010*; *Zimin et al., 2016*). Hence, our results suggest that glutamatergic VN neurons are unable to achieve the energy requirements of increased firing rates. Thus, it is likely that failure of glutamatergic VN projections to the PreBotC allowing appropriate responses to physiological needs underlies the breathing deficits observed in Vglut2:Ndufs4cKO mice.

Development of brainstem lesions correlate with motor deficits in animals constitutively lacking *Ndufs4* (*Quintana et al., 2010*). Accordingly, strategies that improve motor symptoms in Ndufs4KO mice, such as AAV-based gene therapy (*Quintana et al., 2012*; *Di Meo et al., 2017*), rapamycin administration (*Johnson et al., 2013*), or hypoxia (*Jain et al., 2016*; *Ferrari et al., 2017*) cause a marked reduction in brainstem lesions. Here, we have identified a critical role for Vglut2-expressing glutamatergic neurons, in the brainstem and cerebellum, in the development of the motor deficits observed after *Ndufs4* deficiency. In keeping with this, conditional deletion of *Ndufs4* in dopaminergic neurons does not cause cell loss (*Kim et al., 2015*) or motor deficits (*Choi et al., 2017*). However, other areas, such as the striatum, may participate in the delayed, mild, progressive motor dysfunction observed in LS patients (*Chen et al., 2017a*; *Di Meo et al., 2017*).

Our gene expression analysis in brainstem of Vglut2:Ndufs4cKO mice has enabled the generation of an in-depth profile of the transcriptomic landscape in an affected brain area after *Ndufs4* deficiency. This analysis revealed a prominent increase in inflammatory mediators in the affected tissue. However, anti-inflammatory or immunotherapeutic approaches have been mostly ineffective as treatments for LS (*Johnson et al., 2013*; *Finsterer and Zarrouk-Mahjoub, 2017*) with only a few successful cases reported (*Chuquilin et al., 2016*). Our deconvolution data show a marked infiltration of distinct leucocyte populations, underscoring the complex cellular milieu elicited by mitochondrial dysfunction that may underlie the failure of global anti-inflammatory approaches. Delineation of the immune cells recruited to the brain lesions may lead to novel therapeutic approaches tailored for LS.

The *Gad2*-expressing GABAergic neurons do not participate in the appearance of respiratory or motor deficits in *Ndufs4* deficiency. However, they are critical for body temperature control and the onset and development of the fatal epileptic seizures, features that are observed in both global Ndufs4KO mice (*Quintana et al., 2010*) and LS patients (*Finsterer, 2008*; *Koenig, 2008*; *Finsterer and Zarrouk Mahjoub, 2012*). Lack of *Ndufs4* in *Gad2*-expressing neurons leads to the appearance of neuroinflammation in the basal ganglia nuclei such as the GPe and SNr, in agreement with the increased vulnerability of basal ganglia neurons to mitochondrial dysfunction (*Gubellini et al., 2010*). Furthermore, we show that electrophysiological alterations in the GPe neurons predate cortical epileptic events, suggesting a primary role of the basal ganglia circuitry in the development of epileptic seizures in Gad2:Ndufs4cKO mice. Basal ganglia are involved in epilepsy (*Rektor et al., 2012*; *Badawy et al., 2013*; *Vuong and Devergnas, 2018*), likely by acting as an inhibitory input to cortical seizure spread via feedback mechanisms (*Rektor et al., 2012*). Hence, we hypothesize that basal ganglia inhibitory network is affected in Gad2:Ndufs4cKO mice, being unable to control the activity of cortical excitatory neurons, thus leading to epilepsy. Gad2:Ndufs4cKO mice are resistant to different antiepileptic approaches, such as the widely-used antiepileptic drugs carbamazepine, perampanel, and levetiracetam. Although earlier administration of these drugs may have led to a better antiepileptic outcome, epilepsy-induced death in mitochondrial disease patients is usually linked to refractory epileptic seizures (*Finsterer and Zarrouk Mahjoub, 2012*). Hence, Gad2:Ndufs4cKO mice may represent an excellent model to study epileptic mechanisms in LS, a much needed area of research, especially considering that most commonly used antiepileptic drugs may promote mitochondrial toxicity (*Finsterer, 2017*).

As described, both LS patients (*Finsterer, 2008*) and NDUFS4-LS patients (*Ortigoza-Escobar et al., 2016*) present predominant basal ganglia and brainstem affectation. Accordingly, basal ganglia and brainstem lesions are prominent features in global Ndufs4KO (References: *Quintana et al., 2010*; *Quintana et al., 2012*; *Table 2*), and GABAergic and glutamatergic

conditional Ndufs4KO mice, respectively. However, alterations in other areas such as thalamus, cerebellum and spinal cord are also frequently observed, contributing to the clinical complexity of the pathology (*Arii and Tanabe, 2000*; *Lake et al., 2015*). In line with our previous studies (*Quintana et al., 2010*; *Quintana et al., 2012*), here we show the glutamatergic origin of cerebellar and spinal cord alterations, probably contributing motor deficits observed in LS and Ndufs4KO. Clinically, LS patients commonly present hypothermia and failure to thrive (*Finsterer, 2008*), which are recapitulated in global Ndufs4KO mice (*Quintana et al., 2010*). Our work shows reduced body weight and hypophagia in both Vglut2:Ndufs4cKO and Gad2:Ndufs4cKO, while hypothermia is mainly restricted to the latter. Central control of food intake and thermoregulation heavily rely on glutamatergic and GABAergic hypothalamic neuronal populations (*Tan and Knight, 2018*; *Sternson and Eiselt, 2017*; *Zhao et al., 2017*). Hence, our results indicate, that even in the absence of overt neuroinflammation, neuronal *Ndusf4* deficiency may lead to hypothalamic impairment, as observed in LS patients (*Zinka et al., 2010*).

One remaining question is the characterization of the underlying molecular mechanisms leading to the specific vulnerability of defined neuronal populations to *Ndufs4* deficiency. In this regard, increased oxidative stress is one of the hallmarks of the phenotype (*Quintana et al., 2010*). Accordingly, antioxidant treatments have proven moderate effectivity in the global Ndufs4KO mice (*Liu et al., 2015*; *de Haas et al., 2017*). Initiation of extrinsic apoptotic cascades has also been found in Ndufs4KO mice (*Finsterer and Zarrouk-Mahjoub, 2017*, and this work), even though EM imaging demonstrated mostly necrotic death in affected brain regions of Ndufs4KO mice (*El Sabbagh et al., 2010*). In this regard, the remarkable immune cell infiltration described in this work may contribute to the initiation of these cascades. Finally, different studies have pointed at metabolic dysregulation as a potential contributor to the pathology. In this regard, mTOR inhibition or hypoxic conditions modify glycolytic levels, leading to clinical sign amelioration and extended lifespan in Ndufs4KO mice (*Johnson et al., 2013*; *Jain et al., 2016*).

In conclusion, we provide new insights on the genetic identity of affected neuronal populations in LS by dissecting the associated cell type-specific molecular, biochemical, clinical and behavioral features in a model of LS. Our work highlights the importance of addressing mitochondrial disease at the cell type-specific level. The advent of new tools to assess transcriptomic and biochemical changes at this level of resolution (*Sanz et al., 2009*; *Bayraktar et al., 2019*) bodes well for more progress. Hence, our work broadens current understanding of the etiology of LS and paves the way for future studies at the cell type-specific level to unravel the molecular determinants of neuronal pathology in LS.

# Materials and methods

## Key resources table

| Reagent type (species) or resource | Designation | Source or reference | Identifiers | Additional information |
|---|---|---|---|---|
| Genetic reagent (*M. Musculus*) | *Slc17a6*Cre (BAC-Vglut2::Cre) | Borgius L, et al. Mol Cell Neurosci. 2010; 45(3):245–57 | MGI:4881727 | Tg(Slc17a6-icre)1Oki |
| Genetic reagent (*M. Musculus*) | *Gad2*Cre/+ (Gad2-IRES-Cre) | Taniguchi H, et al. Neuron. 2011; 71(6):995–1013. | MGI:4418713 | B6.J.Cg-Gad2tm2 (cre)Zjh/MwarJ JAXs Stock No: 028867 |
| Genetic reagent (*M. Musculus*) | *Chat*Cre/+ (Chat-IRES-Cre) | Rossi J, et al. Cell Metab. 2011; 13(2):195–204 | MGI:6121363 | B6.129S-Chattm1(cre) Lowl/MwarJ JAXs Stock No: 031661 |
| Genetic reagent (*M. Musculus*) | *Ndufs4*Δ/+ | Kruse SE, et al. Cell Metab. 2008; 7 (4):312–20 | MGI:5614215 | B6.129S4-Ndufs4tm1. 1Rpa/J JAXs Stock No: 027058 |
| Genetic reagent (*M. Musculus*) | *Ndufs4*lox/lox | Kruse SE, et al. Cell Metab. 2008; 7 (4):312–20 | MGI:5613135 | B6.129S4-Ndufs4tm1 Rpa/J JAXs Stock No: 026963 |

## Study approval

All experiments were conducted following the recommendations in the Guide for the Care and Use of Laboratory Animals and were approved by the Animal Care and Use Committee of the Seattle Children´s Research Institute and the Universitat Autònoma de Barcelona.

## Animal husbandry

Mice were maintained with Teklad Global rodent diet No. 2014S (HSD Teklad Inc, Madison, Wis.) and water available *ad libitum* in a vivarium with a 12 hr light/dark cycle at 22˚C.

## Mouse genetics

The following mouse lines were used in this study: $Slc17a6^{Cre}$ (BAC-Vglut2::Cre) (*Borgius et al., 2010*) mice were generated by Ole Kiehn. $Gad2^{Cre/+}$ (Gad2-IRES-Cre) (*Taniguchi et al., 2011*) and $Chat^{Cre/+}$ (Chat-IRES-Cre) (*Rossi et al., 2011*) mice were obtained from The Jackson Laboratory (Stock No: 028867 and 031661, respectively) (Bar Harbor, ME). $Ndufs4^{lox/lox}$ and $Ndufs4^{\Delta/+}$ were previously generated by our group (*Quintana et al., 2010*; *Kruse et al., 2008*). Male and female mice of different ages were used in this study. Sex and age of the animals are described in the figure legends. All mice were on a C57BL/6J background after backcrossing for at least 10 generations.

Mice with conditional deletion of $Ndufs4$ in Vglut2-expressing glutamatergic neurons ($Slc17a6^{Cre}$, $Ndufs4^{\Delta/lox}$ or Vglut2:Ndufs4cKO) were generated by crossing mice with one $Ndufs4$ allele deleted and expressing a codon-improved Cre recombinase (iCre) under the $Slc17a6$ promoter ($Slc17a6^{Cre}$, $Ndufs4^{\Delta/+}$) to mice with two floxed $Ndufs4$ alleles ($Ndufs4^{lox/lox}$). Mice with conditional deletion of $Ndusf4$ in $Gad2$-expressing GABAergic neurons ($Gad2^{Cre/+}$, $Ndufs4^{lox/lox}$ or Gad2:Ndufs4cKO) were obtained by crossing mice with one floxed $Ndufs4$ allele and expressing Cre recombinase under the control of the $Gad2$ promoter ($Gad2^{Cre/+}$, $Ndufs4^{lox/+}$) to mice carrying two floxed $Ndufs4$ alleles ($Ndufs4^{lox/lox}$). Similarly, mice with conditional $Ndufs4$ deletion in ChAT-expressing cholinergic neurons ($Chat^{Cre/+}$, $Ndufs4^{lox/lox}$ or ChAT:Ndufs4cKO) were obtained by crossing mice carrying one floxed $Ndufs4$ allele and expressing Cre recombinase driven by the ChAT promoter ($Chat^{Cre/+}$, $Ndufs4^{lox/+}$ mice) to $Ndufs4^{lox/lox}$ mice. Littermate controls were $Slc17a6^{Cre}$, $Ndufs4^{lox/+}$ (Vglut2: Ndufs4cCT); $Gad2^{Cre/+}$, $Ndufs4^{lox/+}$ (Gad2:Ndufs4cCT) and $Chat^{Cre/+}Ndufs4^{lox/+}$ (ChAT:Ndufs4cCT) mice. In all cases, genotype of the offspring and absence of ectopic recombination (i.e. presence of recombination bands in tail DNA samples) was determined by PCR analysis. Primer sequences have been described (*Kruse et al., 2008*).

## Clinical evaluation

Vglut2:Ndufs4cKO and Gad2:Ndufs4cKO mice were examined every other day for clinical signs resulting from cell type-specific $Ndufs4$ inactivation. Physiological (body weight) and behavioral (locomotor activity, motor coordination, gait/postural alterations) parameters were evaluated in more than 50 animals for each mouse line and were grouped into the following categories based on visual observation: '+++" severe manifestation of the clinical sign, '++" moderate manifestation of the clinical sign, '+" mild clinical sign, "- "absence of clinical sign. Mice were humanely euthanized after losing 20% of their peak body weight. Only Vglut2:Ndufs4cKO mice presented the overt and progressive clinical signs. Albeit the presence of individual variability in the development of the disease, early stage was defined in the range P20-P40, mid-stage between P40-P60, and late stage at ages over P60 in Vglut2:Ndufs4cKO.

## Food intake analysis

Food consumption was recorded from 7 to 11 weeks of age using a Physiocage system (Panlab, Spain). Data at 8 weeks of age (right before the median survival value) are presented, including enough individuals to ensure sufficient statistical power.

## Behavioral assays

### Rotarod test

A standard rotarod device (Rotarod, San Diego Instruments, USA) was used to assess motor coordination and global physical condition of animals. Mice were placed on the spindle, which linearly accelerated from 4 to 40 r.p.m and increased 2 r.p.m. every 10 s. Each mouse received five trials per

day over 3 days with a 5 min rest period between trials. The trial ended when the mouse fell off the spindle or after 3 min (cut-off time).

## Open-field

Mice were placed in the open-field arena (560 [W]×365 [D]×400 [H] mm) and allowed to move freely for 10 min. Locomotor activity of mice was next monitored and total distance traveled (m) and velocity (cm/s) measured using the EthoVision tracking software (Noldus).

## Whole-body plethysmography

Ventilatory function was assessed by whole–body plethysmography (EMMS, England, UK) under unrestrained conditions. The system was calibrated to 1 ml volume. An acclimation period of 45 min was allowed for mice adaptation to the chamber, followed by a 15 min experimental period. For Vglut2:Ndufs4cKO mice, plethysmography recordings were performed at different stages of the disease according to the clinical examination, and compared to littermate controls. Studies were also conducted on Gad2:Ndufs4cKO mice between 50–60 days of age and compared to corresponding controls. Respiratory frequency ($F_R$; breaths·minute$^{-1}$) and tidal volume normalized per body weight ($V_T$; µL·g$^{-1}$) were measured.

## Tissue preparation

For immunofluorescence, mouse brains were collected and fixed overnight in 4% paraformaldehyde (PFA) in PBS (pH 7.4). Subsequently, brains were cryoprotected in a PBS solution containing 30% sucrose and frozen in dry ice. Frozen brains were embedded in OCT, sectioned at 30 µm in a cryostat and rinsed in PBS prior to staining. For western blot analysis, brain areas (olfactory bulb, thalamus, spinal cord and globus pallidus) were dissected according to the Paxinos mouse brain atlas (*Paxinos and Frank, 2013*) and flash-frozen in liquid nitrogen before homogenization.

## Immunofluorescence

Tissue sections were rinsed in PBS-0.2% Triton X-100 (PBST) solution. Non-specific binding was blocked with 10% normal donkey serum (NDS) in a PBST solution for 60 min at room temperature, followed by overnight incubation at 4°C with primary antibodies diluted in 1% NDS-PBST (1:2000 for mouse anti-GFAP, Sigma; 1:1000 for chicken anti-GFAP, Abcam; 1:1000 for anti-Iba-1, Wako; 1:1000 for anti-TH, Millipore). Sections were then washed in PBST and incubated for 1 hr at room temperature with the corresponding Cy- (1:200, Jackson Immunoresearch) or Alexa Fluor-conjugated secondary antibodies (1:500, Thermo Fisher Scientific) in 1% NDS-PBST. Sections were finally washed in PBS and rinsed in water before mounting onto slides with Fluoromount G (Electron Microscopy Sciences) for microscopic analysis.

## Immunofluorescence labeling of neuromuscular junctions

Gastrocnemius muscles were sectioned in 60 µm longitudinal sections, collected in 24-well plates in sequential series of 4 slices per well in antifreezing solution. Sections were then blocked with PBS-0.3%Triton-5%Normal Donkey serum and incubated 48 hr at 4°C with primary antibodies anti-synaptophysin (1:500; AB130436, Abcam, UK) and anti-neurofilament 200 (NF200, 1:1000; AB5539, Millipore, USA). After washes, sections were incubated overnight with Alexa 594-conjugated secondary antibody (1:200; A11042, Invitrogen, USA) and Alexa 488 conjugated alfa-bungarotoxin (1:200; B13422, Life Technologies, USA). Slides with the sections were then mounted in Fluoromount-G (Southern Biotech, USA). Confocal images were captured with a scanning confocal microscope (LSM 700 Axio Observer, Carl Zeiss 40x/1.3 Oil DIC M27, Germany). Maximum projections images shown in this study were created from 1.5 µm z projections. For neuromuscular junctions analysis, the proportion of fully occupied endplates was determined by classifying each endplate as fully innervated (when presynaptic terminals overly the endplate), partially innervated (when presynaptic terminals were not clearly within the endplate) or vacant (no presynaptic label in contact with the endplate). At least 3–4 fields with more than 80 endplates were analyzed per each muscle.

## Western blotting

Brain tissue samples were homogenized in iced-cold RIPA buffer (Santa Cruz Biotechnology) and protein concentration determined by the BCA assay (Thermo Fisher Scientific). Thereafter, 20 μg of protein lysates were heat-denatured in Laemmli sample buffer (Bio-Rad Laboratories, Inc), subjected to 4–20% gradient SDS-PAGE and transferred to nitrocellulose membranes (Bio-Rad Laboratories, Inc). Membranes were then blocked for 1 hr with 5% (w/v) dried skimmed milk in Tris-buffered saline containing 0.1% Tween-20 (TBS-T) and incubated overnight at 4°C with primary antibodies against NDUFS4 (Abcam, mouse, 1:500), NSE (Dako, mouse, 1:1,000), GFAP (Sigma, mouse, 1:50,000), Iba1 (Wako, rabbit, 1:10,000), Active (cleaved) caspase 8 (Cell Signaling Technologies, 1:1000), β-actin (Sigma, mouse, 1:20,000) or GAPDH (GeneTex, mouse, 1:40,000). After incubation with the corresponding HRP-conjugated secondary antibodies (1:10,000; Jackson ImmunoResearch), membranes were washed in TBS-T and developed using an enhanced chemiluminescence (ECL) detection system (Pierce). Bands were quantified using Image J software (National Institutes of Health, USA).

## Whole-genome gene expression (WGGEX) analysis

For WGGEX analysis, 150 ng of total RNA extracted from the brainstem of late-stage (over P68) Vglut2:Ndufs4cKO (n = 4) and Vglut2:Ndufs4cCT mice (n = 4) was amplified and biotin-labeled using the Illumina TotalPrep RNA Amplification kit (Ambion). 750 ng of the labeled cRNA was hybridized to MouseRef-8 v2 expression beadchips (Illumina) for 16 hr before washing and analyzing according to the manufacturer's directions. Signal was detected using a BeadArray Reader (Illumina), and data were analyzed for differential expression using the GenomeStudio data analysis software (Illumina). Average normalization, the Illumina custom error model, and multiple testing corrections using the Benjamini and Hochberg false discovery rate were applied to the analysis. Only transcripts with a differential score of >13 (p<0.05) were considered. Normalized and raw data have been deposited in the National Center for Biotechnology Information Gene Expression Omnibus (accession number GSE125470). Functional enrichment analysis of differentially expressed mRNAs (1.4 fold or higher) using overrepresentation analysis (ORA) was accomplished using WebGestalt (http://www.webgestalt.org)(*Wang et al., 2017*). Characterization of the immune cell composition in these gene expression profiles was accomplished using the computational algorithm ImmuCC (*Chen et al., 2017b*).

## Surgery

Mice underwent survival surgery to implant EEG and EMG electrodes under isoflurane anesthesia with subcutaneous bupivacaine (1 mg/kg) for analgesia as described (*Kalume et al., 2013*). Using aseptic technique, a midline incision was made anterior to posterior to expose the cranium. Each EEG electrode consisted of a micro-screw attached to a fine diameter (130 μm bare; 180 μm coated) silver wire. The screw electrodes were placed through the small cranial burr holes at visually identified locations: left and right frontal cortices approximately 1 mm anterior to the bregma and 3 mm lateral of the sagittal suture. EMG electrodes were placed in back muscles. A reference electrode was placed at the midline cerebellum and a ground electrode was placed subcutaneously over the back and the skin was closed with sutures. All electrodes connected to a micro-connector system and their impedances were typically <10 kΩ. After electrode placement, the skin was closed with sutures and the mice were allowed to recover from surgery for 2–3 days.

## Video-EEG-EMG

Recording approach was performed as described (*Kalume et al., 2013*). Simultaneous video-EEG-EMG records were collected in conscious mice on a PowerLab 8/35 data acquisition unit using Labchart 7.3.3 software (AD Instruments, Colorado Spring, Co). All bio-electrical signals were acquired at 1 KHz sampling rate. The EEG signals were processed off-line with a 1–80 Hz bandpass filter and the EMG signals with a 3 Hz highpass filter. Video-EEG-EMG data collected were analyzed using Labchart software.

## Thermal seizure induction

Mouse body temperature was controlled using a rectal temperature probe and a heat lamp attached to a temperature controller in a feedback loop (Physitemp Instruments Inc, NJ). Briefly, as

described (*Oakley et al., 2009*), body temperature was increased by 0.5°C every 2 min until seizure occurred or a 42°C temperature was reached. Mice were immediately cooled using a small fan.

## Electrophysiological studies

### Surgery

Vglut2:Ndufs4cKO and Gad2:Ndufs4cKO mice and their respective controls (30 to 40 days old) were anesthetized with 1.5% isoflurane and implanted with a homemade implant in either the vestibular nuclei (AP: −6.0; ML: −1.0; DV: −4.00) or the lateral globus pallidus (AP: −0.46; ML: −1.95; DV: −4.00), according to Paxinos (*Paxinos and Frank, 2013*). Briefly, a small craniotomy window was made above the desired recording site and a 4-tetrode bundle was lowered into the brain until its destination at 0.1 mm/sec (Robot Stereotaxic, Neurostar, Germany). As ground, a stainless steel wire (0.075 mm diameter, Advent Research Material Ltd., England) was placed between skull and dura over the cerebellum or visual cortex. Then the entire implant was attached and secured to the animal head with dental cement. After surgery, animal welfare and body weight were documented daily and mice allowed to recover for 10–14 days. In Vglut2:Ndufs4cKO mice and their respective controls, a viral vector expressing the light-sensitive cation channel Channelrodopsin-2 in a Cre-dependent manner (AAV-DIO-ChR2) was delivered into the VN prior to implantation of the tetrode to identify glutamatergic Vglut2-expressing cells after blue light (473 nm) stimulation.

### Electrophysiological data acquisition

To explore presence of seizures, the electrophysiological activity was recorded from Gad2:Ndufs4cCT and Gad2:Ndufs4cKO freely-moving mice. After the recovery period, animals were placed on a daily basis in an open-field arena and both local field potentials (LFPs) and extracellular single-unit activity were recorded until the death of the animal. In Vglut2:Ndufs4cCT and Vglut2:Ndufs4cKO mice, a tetrode bundle was attached along a fiber optic (Ø200 μm Core, 0.22 NA, FG200UCC, Thorlabs, USA) to deliver light and record neuronal cells at the same time. Again, all animals were daily recorded in an open-field until death.

For all experiments, local field potential activity was amplified, A-D converted and sampled at 1 KHz and bandpass filtered at 0.1 to 250 Hz (DigitalLynx SX and Cheetah Data Acquisition System, Neuralynx, USA). Continuous spike signals were also recorded, amplified, band-pass filtered (300 Hz to 8 kHz) and sampled at 32 kHz. Optogenetic stimulations (ranging from 50 to 200 ms blue light stimuli, 473 nm, delivered frequency from 1 to 40 Hz) were delivered through a blue-emitting diode laser (473 nm DPSS Laser System, Laserglow Technologies, CA).

### Electrophysiological data analysis

For the presence of seizures, data were analyzed in Spike2 (Cambridge Electronic Design Limited, UK) for visual inspection and report presence of epileptic events. Offline single-unit spike sorting was performed with Offline Sorter software (Offline Sorter, Plexon Inc, USA). Briefly, for each channel, a specific manual threshold was defined and all events bypassing this threshold were assumed to be an action potential. After an overall waveform shape and three principal component analysis (PCA) inspection, all spikes were sorted with an automatic K-mean algorithm to separate clusters of cells. After sorting, a final inspection of ISI histogram and sorting statistics (MANOVA F statistics, J3 and the Davies-Bouldin validity index) was performed to ensure the best single-unit clustering. In Vglut2:Ndufs4cCT and Vglut2:Ndufs4cKO mice, to identify glutamatergic neurons from all sorted cells, response to optogenetic stimulations were plotted. For all stimulation trains, during a baseline activity corresponding to 500 ms before stimulation, the mean number of spikes per bins was calculated. Then, a single cell was identified as glutamatergic only if, during optogenetic stimulation, there were bins expressing a number of spikes superior to the baseline mean plus three times the standard deviation of the baseline mean. Finally, after analysis of spiking activity (Matlab, MathWorks, USA), cells with a coefficient of variation of interspike interval (CV.ISI) over three were exclude from the data set.

### Body core temperature measurement

The baseline temperature was acquired as follows: As for thermal seizure induction (*Oakley et al., 2009*), a T-type implantable rectal probe was placed and taped to the tail with a lab tape. The probe

was connected to a temperature monitoring device (T-pod) and the resting body core temperature was monitored in unrestrained mice using a PowerLab 8/35 data acquisition unit and Labchart 7.3.3 software (AD Instruments, Colorado Spring, Co) for 10 min. The telemetric data are used for long-term monitoring of temperature and its oscillation during circadian rhythm and sleep. Note that baseline data obtained via the two methods are consistent.

### Drug treatments

Levetiracetam (Keppra parental formulation), perampanel (Clinisciences) and carbamazepine (Sigma-Aldrich) antiepileptic drugs were injected intraperitoneally in Gad2:Ndufs4cKO mice daily, starting at P40. Levetiracetam and perampanel were dissolved in saline solution and injected at a dose of 60 mg/kg and 0.75 mg/kg, respectively. Carbamazepine was slowly diluted in PEG300, dissolved in saline, and injected at 40 mg/kg. A control group for each drug was obtained by injecting each respective vehicle to Gad2:Ndufs4cKO mice.

### qRT-PCR analysis

qRT-PCR assays were performed as described (*Sanz et al., 2015*). Briefly, equal amounts of RNA were assayed using the Power SYBR green RNA-to-Ct 1-Step Master Mix (Applied Biosystems) or the TaqMan RNA-to-Ct 1-Step Master Mix (Applied Biosystems), depending on the system used (SYBR or Taqman). Relative expression values were obtained using the standard curve method and normalized to *Actb* levels. Amplification efficiencies were calculated using the AriaMx software (Agilent) and were within accepted parameters (80–120%). *Gad2* mRNA was determined using a specific Taqman assay (Mm00484623_m1; Applied Biosystems), and sequences for the different primer sets used in SYBR assays (*Slc17a6* and *Actb*) were obtained from Primerbank (*Spandidos et al., 2010*).

### Statistics

Data are shown as the mean ± SEM. GraphPad Prism v5.0 software was used for statistical analyses. Appropriate tests were selected depending on the experimental design as stated in the figure legends. Statistical significance, when reached (p<0.05 was considered significant), is stated in the figure legends.

## Acknowledgements

We thank Richard Palmiter, Diane Durnam and members of the Quintana lab for insightful discussions, comments and edits. We thank our funders for their support.

## Additional information

### Funding

| Funder | Grant reference number | Author |
| --- | --- | --- |
| Ministerio de Economía y Competitividad | JCI-2015-24576 | Irene Bolea |
| European Commission | H2020-MSCA-COFUND-2014-665919 | Alejandro Gella |
| European Commission | H2020-MSCA-IF-2014-658352 | Elisenda Sanz |
| Ministerio de ciencia, investigación y universidades | RTI2018-101838-J-I00 | Elisenda Sanz |
| Ministerio de Economía y Competitividad | BES-2015-073041 | Patricia Prada-Dacasa |
| CIBERNED | CB06/05/1105 | Xavier Navarro |
| TERCEL | RD16/0011/0014 | Xavier Navarro |
| Instituto de Salud Carlos III | | Xavier Navarro |

| | | |
|---|---|---|
| European Regional Development Fund | | Xavier Navarro |
| Seattle Children's Research Institute | Seed Funds | Franck Kalume |
| NINDS | R01 NIH/NS 102796 | Franck Kalume |
| University of Washington Neurological Surgery Department | Ellenbogen Neurological Surgery Research Fund | Franck Kalume |
| University of Washington | The Ryan J. Murphy SUDEP Research Funds | Franck Kalume |
| Seattle Children's Research Institute | Seed Funds | Albert Quintana |
| Mitochondrial Research Guild | Seed Funds | Albert Quintana |
| Ministerio de Economía y Competitividad | RyC-2012-1187 | Albert Quintana |
| European Research Council | ERC-2014-StG-638106 | Albert Quintana |
| Ministerio de Economía y Competitividad | SAF2014-57981P | Albert Quintana |
| Ministerio de Economía y Competitividad | SAF2017-88108-R | Albert Quintana |
| Agència de Gestió d'Ajuts Universitaris i de Recerca | 2017SGR- 323 | Albert Quintana |

The funders had no role in study design, data collection and interpretation, or the decision to submit the work for publication.

**Author contributions**

Irene Bolea, Conceptualization, Data curation, Formal analysis, Funding acquisition, Validation, Investigation, Visualization, Methodology, Writing—original draft, Writing—review and editing; Alejandro Gella, Data curation, Formal analysis, Funding acquisition, Validation, Investigation, Visualization, Methodology, Writing—review and editing; Elisenda Sanz, Data curation, Formal analysis, Supervision, Funding acquisition, Validation, Investigation, Visualization, Methodology, Writing—original draft, Writing—review and editing; Patricia Prada-Dacasa, Funding acquisition, Investigation, Methodology, Writing—review and editing; Fabien Menardy, Formal analysis, Investigation, Visualization, Methodology, Writing—original draft, Writing—review and editing; Angela M Bard, Data curation, Formal analysis, Investigation, Methodology, Writing—original draft, Writing—review and editing; Pablo Machuca-Márquez, Guillem Mòdol-Caballero, Formal analysis, Investigation, Methodology, Writing—review and editing; Abel Eraso-Pichot, Formal analysis, Investigation, Methodology; Xavier Navarro, Resources, Formal analysis, Supervision, Funding acquisition; Franck Kalume, Conceptualization, Data curation, Formal analysis, Funding Acquisition, Validation, Investigation, Visualization, Methodology, Writing—original draft, Writing—review and editing; Albert Quintana, Conceptualization, Resources, Data curation, Formal analysis, Supervision, Funding acquisition, Validation, Investigation, Visualization, Methodology, Writing—original draft, Project administration, Writing—review and editing

**Author ORCIDs**

Irene Bolea ⓘD https://orcid.org/0000-0001-9591-980X
Alejandro Gella ⓘD https://orcid.org/0000-0002-3983-1392
Elisenda Sanz ⓘD http://orcid.org/0000-0002-7932-8556
Patricia Prada-Dacasa ⓘD https://orcid.org/0000-0003-0689-9072
Fabien Menardy ⓘD https://orcid.org/0000-0002-8712-1344
Pablo Machuca-Márquez ⓘD http://orcid.org/0000-0002-7980-3839
Abel Eraso-Pichot ⓘD https://orcid.org/0000-0001-6837-2714
Xavier Navarro ⓘD https://orcid.org/0000-0001-9849-902X

Franck Kalume https://orcid.org/0000-0002-5528-2565
Albert Quintana https://orcid.org/0000-0003-1674-7160

## Ethics

Animal experimentation: All experiments were conducted following the recommendations in the Guide for the Care and Use of Laboratory Animals and were approved by the Animal Care and Use Committee of the Seattle Children´s Research Institute (#00108) and Universitat Autònoma de Barcelona (CEEAH 2925, 3295, 4114, 4155). All surgeries were performed under anesthesia, and every effort was made to minimize suffering.

## Decision letter and Author response

Decision letter https://doi.org/10.7554/eLife.47163.027
Author response https://doi.org/10.7554/eLife.47163.028

# Additional files

## Supplementary files

• Supplementary file 1. Statistical parameters. (A) Statistical parameters for two-way ANOVA. (B) Statistical parameters for two-tailed unpaired t-Tests. (C) Statistical parameters for one-way ANOVA. (D) Statistical parameters for Log-rank (Mantel-Cox) tests.
DOI: https://doi.org/10.7554/eLife.47163.022

• Transparent reporting form
DOI: https://doi.org/10.7554/eLife.47163.023

## Data availability

Normalized and raw data have been deposited in the National Center for Biotechnology Information Gene Expression Omnibus (accession number GSE125470).

The following dataset was generated:

| Author(s) | Year | Dataset title | Dataset URL | Database and Identifier |
|---|---|---|---|---|
| Sanz E, Quintana A | 2019 | Gene expression analysis in the Brainstem of Vglut2:Ndufs4cKO mice | https://www.ncbi.nlm.nih.gov/geo/query/acc.cgi?acc=GSE125470 | NCBI Gene Expression Omnibus, GSE125470 |

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
