## [Decision Letter]

Thank you for submitting your article "Defined neuronal populations drive fatal phenotype in Leigh Syndrome" for consideration by *eLife*. Your article has been reviewed by three peer reviewers, including Matt Kaeberlein as the Reviewing Editor and Reviewer #1, and the evaluation has been overseen by Huda Zoghbi as the Senior Editor. The following individuals involved in review of your submission have agreed to reveal their identity: Isha Jain (Reviewer #2).

The reviewers have discussed the reviews with one another and the Reviewing Editor has drafted this decision to help you prepare a revised submission.

Summary:

This is an interesting and relatively comprehensive characterization of two cell-type specific knockout mouse models lacking the Complex I factor Ndufs4 in either glutamatergic (VGLUT) or GABAergic (Gad2) neurons. The authors convincingly show that VGLUT2 KO causes a disease phenotypically similar to whole body knockout. The Gad2 KO causes a distinct pathology with death largely due to seizures. Although the experiments appear to be generally well conducted, there are some concerns with experimental design and presentation that should be addressed in a revised manuscript.

Essential revisions:

1) There are major concerns about the drug treatment experiments. P40 is likely too late to see significant rescue. It has been shown that rapamycin treatment is greatly attenuated when started at P35 compared to P10, and the time between P35 to P40 is when disease symptoms and neurodegeneration become pronounced. With respect to rapamycin, the interpretation is very likely to be incorrect, since rapamycin initiated at P10 rescues the whole body KO far beyond the lifespan of the Gad2 KO on rapamycin initiated at P40. It seems unlikely that this would be the case if rapamycin initiated at P10 weren't also rescuing the defects in the GABAergic neurons. Optimally, these experiments should be repeated starting treatment earlier in life, before the onset of disease symptoms. If that is not feasible, we would recommend removing the rapamycin experiment and discussing the limited interpretability of the seizure drugs based on the experimental design.

2) Claims are made that cell-type specific loss of Ndufs4 gives rise to regionally-specific inflammation/pathology. Evidence showing the corresponding unaffected regions in each strain (e.g. VN/IO/FN in Gad2 mice and OB, etc. in Vglut2 mice) would be important to support these assertions. Along these lines it would also be nice to have some data (even in supplement) for whole-brain pathology (e.g. histology or whole-brain MRI, showing affected and unaffected regions for both strains).

3) The authors report neuroinflammation that they describe as "reminiscent of the lesions" seen in whole body KO mice, but apparently do not actually identify lesions or neuron loss. The lack of overt lesions seen in the brainstem of these mice could be due to temporal differences between pathological progression in these conditional mice in comparison to whole-body knockout, or they could indicate different mechanisms of pathogenesis, with NDUFS4 loss affecting neuronal function in glutamatergic neurons without overt neuronal loss. Assessing the affected brain regions for early apoptotic signaling seen in the whole-body knockouts may offer clues as to mechanism for the phenotypes observed.

4) It is stated that "premature death was preceded by weight loss", however, it is unclear from the way the data is presented that this is actually the case. Furthermore, temporal context is addressed somewhat inconsistently throughout the manuscript. There are places where the age of the animals at the time of sampling is not explicitly stated, or is unclear – most notably for the findings looking at neuroinflammation and immune responses. The samples used for the transcriptional profiling and the samples in which they identified neuroinflammation are both lacking explicitly stated ages. The manuscript should be carefully edited to ensure that this information is clearly presented for all experiments.

5) Throughout the section entitled "Vglut2:Ndufs4cKO and Gad2:Ndufs4cKO mice manifest clinically distinct phenotypes" the authors reference both data that is not shown, data in the supplemental figures and data that is shown in later figures but is not referenced. Impact of these statements would be increased with the appropriate data quantified and cited. For example, the statement that "these mice showed increased tremor, were completely docile and hypotonic and would not explore new environments" – there is no quantification or data presented to assess the increased tremor, docility, or hypotonia, however, the lack of exploration of new environments is presented in Figure 3C.

6) The statement about "immunosuppression and infection may underlie the eventual demise" of rapamycin treated mice should be removed. Aside from the fact that this is pure speculation with no evidence to support it, it propagates the myth that rapamycin is a true immunosuppressant, despite accumulating evidence that rapamycin can boost immune function, at least in certain contexts. Likewise, the statement about "rapamycin-resistant phenotypes, such as SUDEP" should be removed, given the problems with the experimental design mentioned in #1.

7) The Title needs to be changed. This is a mouse model of Leigh Syndrome and needs to be stated as such. The title currently implies that these findings are true for Leigh Syndrome in patients, which may or may not be the case.

8) Typically, Leigh syndrome patients are described as having "relative neuronal sparing"; It would be good to include discussion of role of astrocytes/glia and how this fits in with your results (cell autonomous vs. non-cell autonomous phenotypes). Are the cells where Ndufs4 is knocked-out, the ones that are specifically dying?

9) While not the focus of this paper, it would be nice to have a description of the potential molecular mechanisms for why these neuronal populations might be specifically affected by loss of Ndufs4 (as opposed to cholinergic neurons, etc.).

10) What happens to body temperature phenotype in these different strains?

---

## [Author Response]

Summary:This is an interesting and relatively comprehensive characterization of two cell-type specific knockout mouse models lacking the Complex I factor Ndufs4 in either glutamatergic (VGLUT) or GABAergic (Gad2) neurons. The authors convincingly show that VGLUT2 KO causes a disease phenotypically similar to whole body knockout. The Gad2 KO causes a distinct pathology with death largely due to seizures. Although the experiments appear to be generally well conducted, there are some concerns with experimental design and presentation that should be addressed in a revised manuscript.

We would like to thank reviewers and editors for the positive words on our work and the detailed and constructive comments on our previous submission, which have improved the overall quality of the manuscript. In this resubmission, we believe that we have successfully addressed the concerns raised by editors and reviewers.

Essential revisions:1) There are major concerns about the drug treatment experiments. P40 is likely too late to see significant rescue. It has been shown that rapamycin treatment is greatly attenuated when started at P35 compared to P10, and the time between P35 to P40 is when disease symptoms and neurodegeneration become pronounced. With respect to rapamycin, the interpretation is very likely to be incorrect, since rapamycin initiated at P10 rescues the whole body KO far beyond the lifespan of the Gad2 KO on rapamycin initiated at P40. It seems unlikely that this would be the case if rapamycin initiated at P10 weren't also rescuing the defects in the GABAergic neurons. Optimally, these experiments should be repeated starting treatment earlier in life, before the onset of disease symptoms. If that is not feasible, we would recommend removing the rapamycin experiment and discussing the limited interpretability of the seizure drugs based on the experimental design.

We agree with the editor and reviewers that in light with previous studies, starting drug treatments at P40 is probably too late to achieve significant rescue. In this regard, similar to the rest of antiepileptic drug administration, the rapamycin experiment was aimed at evaluating its potential activity blocking seizures through mTOR inhibition (Ehninger et al., 2008; Krueger et al., 2013), rather than at improving the neuroinflammatory status of the mice. Hence, the rationale was to provide a treatment coincident (or close to) the onset of epilepsy. In this regard, we have added a new antiepileptic drug treatment (perampanel, an AMPA blocker), also showing no effect on survival. While we agree that earlier administration would have likely had a larger impact on lifespan, in particular with rapamycin, with our current animal protocol it is not feasible to start any treatment significantly earlier. Hence, we have followed editor and reviewer’s advice and removed the rapamycin experiment (Figure 7—figure supplement 2) and discussed the aforementioned limitations in the revised text.

2) Claims are made that cell-type specific loss of Ndufs4 gives rise to regionally-specific inflammation/pathology. Evidence showing the corresponding unaffected regions in each strain (e.g. VN/IO/FN in Gad2 mice and OB, etc. in Vglut2 mice) would be important to support these assertions. Along these lines it would also be nice to have some data (even in supplement) for whole-brain pathology (e.g. histology or whole-brain MRI, showing affected and unaffected regions for both strains).

We have added whole-brain histological data on neuroinflammatory status (assessed by GFAP and IBA1 stainings) for both Gad2 and Vglut2 mice as well as the corresponding unaffected regions in each strain in the revised manuscript, as suggested. This data is now part of Figure 2A-B and Figure 2—figure supplement 1.

3) The authors report neuroinflammation that they describe as "reminiscent of the lesions" seen in whole body KO mice, but apparently do not actually identify lesions or neuron loss. The lack of overt lesions seen in the brainstem of these mice could be due to temporal differences between pathological progression in these conditional mice in comparison to whole-body knockout, or they could indicate different mechanisms of pathogenesis, with NDUFS4 loss affecting neuronal function in glutamatergic neurons without overt neuronal loss. Assessing the affected brain regions for early apoptotic signaling seen in the whole-body knockouts may offer clues as to mechanism for the phenotypes observed.

We agree with editor and reviewers’ comments that “reminiscent of the lesions” may suggest that there are no lesions being developed. We have indeed observed the presence of marked brainstem lesions in some late-stage animals, similar to whole-body knockout, as can be seen in the revised Figure 2, but appearing at later time points in Vglut2:Ndufs4cKO. Hence, this data points at slight temporal differences in the progression between Vglut2:Ndufs4cKO and whole-body Ndufs4KO. Furthermore, while it is possible that restricting *Ndufs4* deficiency to glutamatergic neurons may trigger different mechanisms, following reviewers’ suggestion we show increased levels of active caspase 8 activation in the brainstem of Vglut2:Ndufs4cKO, as has been shown for whole-body Ndufs4KO (Quintana et al., 2010). These data are included in the revised manuscript and the text has been modified accordingly.

4) It is stated that "premature death was preceded by weight loss", however, it is unclear from the way the data is presented that this is actually the case. Furthermore, temporal context is addressed somewhat inconsistently throughout the manuscript. There are places where the age of the animals at the time of sampling is not explicitly stated, or is unclear – most notably for the findings looking at neuroinflammation and immune responses. The samples used for the transcriptional profiling and the samples in which they identified neuroinflammation are both lacking explicitly stated ages. The manuscript should be carefully edited to ensure that this information is clearly presented for all experiments.

We agree with reviewers’ that rather than a weight loss, both mouse lines’ body weights plateau 2-3 weeks before death. This statement has been corrected in the revised manuscript. Furthermore, we have edited the text throughout the manuscript to explicitly state the age of the mice. We apologize for the missing information and /or lack of clarity in the previous version of the manuscript.

5) Throughout the section entitled "Vglut2:Ndufs4cKO and Gad2:Ndufs4cKO mice manifest clinically distinct phenotypes" the authors reference both data that is not shown, data in the supplemental figures and data that is shown in later figures but is not referenced. Impact of these statements would be increased with the appropriate data quantified and cited. For example, the statement that "these mice showed increased tremor, were completely docile and hypotonic and would not explore new environments" – there is no quantification or data presented to assess the increased tremor, docility, or hypotonia, however, the lack of exploration of new environments is presented in Figure 3C.

We thank reviewers for pointing out the missing citations. Indeed, we now adequately cite the statements in Table 1 and Figure 3C, as stated. Furthermore, following reviewers’ comments, we have quantified the presence of ataxia and clasping in Vglut2:Ndufs4KO. These data are described in Figure 1—figure supplement 3.

6) The statement about "immunosuppression and infection may underlie the eventual demise" of rapamycin treated mice should be removed. Aside from the fact that this is pure speculation with no evidence to support it, it propagates the myth that rapamycin is a true immunosuppressant, despite accumulating evidence that rapamycin can boost immune function, at least in certain contexts. Likewise, the statement about "rapamycin-resistant phenotypes, such as SUDEP" should be removed, given the problems with the experimental design mentioned in #1.

We have removed these sentences, as suggested.

7) The Title needs to be changed. This is a mouse model of Leigh Syndrome and needs to be stated as such. The title currently implies that these findings are true for Leigh Syndrome in patients, which may or may not be the case.

The Title has been changed to imply this study has been performed in a mouse model of Leigh Syndrome.

8) Typically, Leigh syndrome patients are described as having "relative neuronal sparing"; It would be good to include discussion of role of astrocytes/glia and how this fits in with your results (cell autonomous vs. non-cell autonomous phenotypes). Are the cells where Ndufs4 is knocked-out, the ones that are specifically dying?

In agreement with the reviewer, our previous work has shown that an astrocyte-neuron crosstalk is necessary to induce neurodegeneration in the context of mitochondrial disease (Liu et al., 2015), and it is now discussed in more depth in the revised manuscript. Furthermore, in our model, neuronal loss is mostly evident in late stage whole-body Ndufs4KO mice (Quintana et al., 2010). As discussed in point #3, neuroinflammation and lesions appear less dramatic in conditional Ndufs4KO mice, which could point to reduced neurodegeneration. Accordingly, as a proxy of cell type-specific death, we now show by qRT-PCR analysis no significant differences in the levels of Vlgut2 (*Slc17a6*) or *Gad2* transcripts in relevant brain areas for each conditional Ndufs4KO line (Figure 2G-H).

9) While not the focus of this paper, it would be nice to have a description of the potential molecular mechanisms for why these neuronal populations might be specifically affected by loss of Ndufs4 (as opposed to cholinergic neurons, etc.).

We agree with reviewer that this interesting point is beyond the scope of the current manuscript, and it is an active of research in the lab. While the potential differential mechanisms have not been elucidated yet, we have included a discussion on the particularities of glutamatergic and GABAergic neurons that may underlie this enhanced susceptibility.

10) What happens to body temperature phenotype in these different strains?

We thank reviewers for raising this point, we have identified that GABAergic neurons are involved in the body temperature phenotype observed in Ndufs4KO mice. Gad2:Ndus4cKO mice present an early affectation of body temperature control, with reduced body temperatures at all tested ages. On the other hand, Vglut2:Ndufs4cKO only show a reduction in body temperature in a mid-late stage of the disease. These results are shown in Figure 6 in the revised manuscript.